# Dynamic Regret Analysis of Safe Distributed Online Optimization for Convex and Non-convex Problems

**Ting-Jui Chang**                                                                 *chang.tin@northeastern.edu*
*Department of Mechanical and Industrial Engineering*
*Northeastern University*

**Sapana Chaudhary**                                                                 *sapanac@tamu.edu*
*Department of Electrical and Computer Engineering*
*Texas A&M University*

**Dileep Kalathil**                                                                 *dileep.kalathil@tamu.edu*
*Department of Electrical and Computer Engineering*
*Texas A&M University*

**Shahin Shahrampour**                                                                 *s.shahrampour@northeastern.edu*
*Department of Mechanical and Industrial Engineering*
*Northeastern University*

**Reviewed on OpenReview:** *https://openreview.net/forum?id=xiQXHvL1eN*

## Abstract

This paper addresses safe distributed online optimization over an unknown set of linear safety constraints. A network of agents aims at jointly minimizing a *global*, *time-varying* function, which is only partially observable to each individual agent. Therefore, agents must engage in *local* communications to generate a *safe* sequence of actions competitive with the best minimizer sequence in hindsight, and the gap between the performance of two sequences is quantified via dynamic regret. We propose distributed safe online gradient descent (D-Safe-OGD) with an exploration phase, where all agents estimate the constraint parameters collaboratively to build estimated feasible sets, ensuring the action selection safety during the optimization phase. We prove that for convex functions, D-Safe-OGD achieves a dynamic regret bound of $O(T^{2/3}\sqrt{\log T} + T^{1/3}C_T^*)$, where $C_T^*$ denotes the path-length of the best minimizer sequence. We further prove a dynamic regret bound of $O(T^{2/3}\sqrt{\log T} + T^{2/3}C_T^*)$ for certain non-convex problems, which establishes the first dynamic regret bound for a *safe distributed* algorithm in the *non-convex* setting.

## 1 Introduction

Online learning (or optimization) is a sequential decision-making problem modeling a repeated game between a learner and an adversary (Hazan, 2016). At each round $t$, $t \in [T] \triangleq \{1, \ldots, T\}$, the learner chooses an action $\mathbf{x}_t$ in a convex set $\mathcal{X} \subseteq \mathrm{R}^d$ using the information from previous observations and suffers the loss $f_t(\mathbf{x}_t)$, where the function $f_t : \mathcal{X} \to \mathrm{R}$ is chosen by the adversary. Due to the sequential nature of the problem, a commonly used performance metric is *regret*, defined as the difference between the cumulative loss incurred by the learner and that of a benchmark comparator sequence. When the comparator sequence is fixed, this metric is called *static* regret, defined as follows

$$\mathbf{Reg}_T^s \triangleq \sum_{t=1}^T f_t(\mathbf{x}_t) - \min_{\mathbf{x} \in \mathcal{X}} \sum_{t=1}^T f_t(\mathbf{x}). \tag{1}$$

Static regret is well-studied in the online optimization literature. In particular, it is well-known that online gradient descent (OGD) achieves an $O(\sqrt{T})$ (respectively, $O(\log T)$) static regret bound for convex (respectively, exp-concave

and strongly convex) problems (Zinkevich, 2003; Hazan et al., 2007), and these bounds are optimal in the sense of matching the lower bound of regret in the respective problems (Hazan, 2016).

For non-convex problems, however, we expect that the standard regret notion used in the convex setting may not be a tractable measure for gauging the algorithm performance. For example, in the context of online non-convex optimization, Hazan et al. (2017) quantified the regret in terms of the norm of (projected) gradients, consistent with the stationarity measure in offline optimization. More recently, Ghai et al. (2022) showed that under certain geometric and smoothness conditions, OGD applied to non-convex functions is an approximation of online mirror descent (OMD) applied to convex functions under a reparameterization. In view of this equivalence, OGD achieves an $O(T^{2/3})$ static regret that is defined in (1).

A more stringent benchmark for measuring the performance of online optimization algorithms is the *dynamic* regret (Besbes et al., 2015; Jadbabaie et al., 2015), defined as

$$\mathbf{Reg}_T^d \triangleq \sum_{t=1}^{T} f_t(\mathbf{x}_t) - \sum_{t=1}^{T} f_t(\mathbf{x}_t^*), \tag{2}$$

where $\mathbf{x}_t^* \triangleq \operatorname{argmin}_{\mathbf{x} \in \mathcal{X}} f_t(\mathbf{x})$. It is well-known that dynamic regret scales linearly with $T$ in the worst-case scenario, when the function sequence fluctuates drastically over time. Therefore, various works have adopted a number of variation measures to characterize the dynamic regret bound. We provide a detailed review of these measures in Section 2 and describe the *safe distributed* online optimization problem (which is the focus of this work) in the next section.

## 1.1 Safe Distributed Online Optimization

There are two distinctive components that make "safe distributed online optimization" more challenging than the standard centralized online optimization:

(i) **Distributed Setting:** Distributed online optimization has been widely applied to robot formation control (Dixit et al., 2019b), distributed target tracking (Shahrampour & Jadbabaie, 2017), and localization in sensor networks (Akbari et al., 2015). In this setting, a network of $m$ agents aims at solving the online optimization problem collectively. The main challenge is that the time-varying function sequence is only partially observable to each individual agent. Each agent $j \in [m]$ receives (gradient) information about the "local" function $f_{j,t}(\cdot)$, but the objective is to control the dynamic regret of each agent with respect to the global function $f_t(\cdot) = \sum_{i=1}^{m} f_{i,t}(\cdot)$, i.e.,

$$\mathbf{Reg}_{j,T}^d \triangleq \sum_{t=1}^{T} f_t(\mathbf{x}_{j,t}) - \sum_{t=1}^{T} f_t(\mathbf{x}_t^*) = \sum_{t=1}^{T} \sum_{i=1}^{m} f_{i,t}(\mathbf{x}_{j,t}) - \sum_{t=1}^{T} f_t(\mathbf{x}_t^*). \tag{3}$$

Therefore, despite availability of only local information, the action sequence of agent $j$ is evaluated in the global function and is compared to the global minimizer sequence. It is evident that agents must communicate with each other (subject to a graph/network constraint) to approximate the global function, which is common to distributed problems. The discussion on the network structure and communication protocols is provided in Section 3.3.

(ii) **Safety Constraints:** The literature on distributed online optimization has mostly focused on problems where the constraint set $\mathcal{X}$ is known, and less attention has been given to problems with *unknown* feasible sets (see Section 2 for a comprehensive literature review). However, in many real-world applications, this set represents certain safety constraints that are *unknown* in advance. Examples include voltage regulation constraints in power systems (Dobbe et al., 2020), transmission bandwidth in communication networks due to human safety considerations (Luong et al., 2019) and stabilizing action set in robotics applications (Åström & Murray, 2010). In these scenarios, one needs to perform parameter estimation to *learn* the safety constraints while ensuring that the regret is as small as possible.

## 1.2 Contributions

In this work, we address the problem of *distributed* online optimization with *unknown* linear safety constraints. In particular, the constraint set

$$\mathcal{X}^s \triangleq \{\mathbf{x} \in \mathbb{R}^d : \mathbf{A}\mathbf{x} \leq \mathbf{b}\},$$

is linear, where $\mathbf{A}$ is *unknown* and must be learned by agents to subsequently choose their actions from this set. The superscript $s$ in $\mathcal{X}^s$ alludes to safety. Our specific objective is to study *dynamic* regret (2) for both convex and non-convex problems when the set $\mathcal{X}^s$ is unknown to agents. Our contributions are three-fold:

1) We propose and analyze safe distributed online gradient descent (D-Safe-OGD) algorithm, which has two phases (exploration and optimization). In the exploration phase, agents individually explore and jointly estimate the constraint parameters in a distributed fashion. Then, each agent constructs a feasible set with its own estimate, which ensures the action selection safety with high probability (Lemma 2). Since the estimates are only local, in the optimization phase, agents apply distributed OGD projected to *different* feasible sets, which brings forward an additional technical challenge. We tackle this using the geometric property of linear constraints (Fereydounian et al., 2020) as well as the sensitivity analysis of perturbed optimization problems with second order regular constraints (Bonnans et al., 1998), which allows us to quantify the distance between projections of a point to two different sets that are "close enough" to each other (Lemma 12).

2) We analyze D-Safe-OGD in the *convex* setting. Due to the challenge discussed in the previous item, we cannot directly apply existing results on distributed online optimization with a common feasible set. The agents must use the exploration time to estimate their own feasible sets, during which they incur linear regret. Therefore, after striking a trade-off between the exploration and optimization phases, we prove (Theorem 3) a dynamic regret bound of $O(T^{2/3}\sqrt{\log T} + T^{1/3}C_T^*)$, where

$$C_T^* \triangleq \sum_{t=2}^{T} \left\| \mathbf{x}_t^* - \mathbf{x}_{t-1}^* \right\|, \tag{4}$$

is the *path-length*, defined with respect to the global minimizer sequence (Mokhtari et al., 2016; Jadbabaie et al., 2015). If the problem is centralized (single agent) and the comparator sequence is fixed, i.e., $\mathbf{x}_t^* = \mathbf{x}$, our result recovers the static regret bound of Chaudhary & Kalathil (2022).

3) We further analyze D-Safe-OGD in the *non-convex* setting. We draw upon the idea of the algorithmic equivalence between OGD and OMD (Ghai et al., 2022) to establish that in certain problem geometries (Assumptions 6-9), D-Safe-OGD can be approximated by a distributed OMD algorithm applied to a reparameterized "convex" problem. We prove that the dynamic regret is upper bounded by $O(T^{2/3}\sqrt{\log T} + T^{2/3}C_T^*)$ in Theorem 5, which is the first dynamic regret bound for a *safe distributed* algorithm in the *non-convex* setting. If the problem is centralized (single agent) and the comparator sequence is fixed, i.e., $\mathbf{x}_t^* = \mathbf{x}$, our result recovers the static regret bound of Ghai et al. (2022) (disregarding log factors).

## 1.3 Summary of the Technical Analysis

We now elaborate on the technical contribution of this work:

1) **Estimation Error**
   To learn the constraint parameters in a distributed way, in the early stage of D-Safe-OGD (i.e., Algorithm 1), agents apply random actions to collectively form a least squares (LS) problem. Then, a distributed optimization algorithm is applied to jointly learn the constraint parameters. Though we apply EXTRA (Shi et al., 2015) in this work, we note that this part can be replaced by other distributed optimization methods suitable for smooth, strongly convex problems. Based on existing results on LS estimators (Theorem 7), matrix concentration inequalities for random matrices (Theorem 8), and the convergence rate of EXTRA (Theorem 10), we quantify the estimation error bound for each agent in Lemma 2.

2) **Gap between Projections on Two Estimated Feasible Sets**
   As mentioned earlier, since the estimated feasible sets are different across agents, in our technical analysis we need to quantify the gap between projections on different estimated sets to derive the final regret bounds. To tackle this, in Lemma 12 we show that if we consider the projection problem for one agent as a quadratic programming with second-order cone constraints, the projection problem for any other agent can be cast as a perturbed version of the first problem. Then, we build on the sensitivity analysis of optimization problems (Theorem 13) to show that the difference between the projected points (optimal solutions of the original

and perturbed problems) is of the same order as the difference between agents estimates of the constraint parameters.

3) **Regret Bounds for Convex and Non-convex Cases**
Building on the previous points, we extend the analysis of the distributed OGD in Yan et al. (2012) to the safe setup (Lemma 14), from which we derive dynamic regret bounds for both convex and non-convex cases. We note that for the non-convex (centralized) case, Ghai et al. (2022) showed that if certain geometric properties are satisfied, OGD on non-convex costs can be well-approximated as OMD on convex costs through reparameterization (Lemma 15). We establish in Lemma 17 that this approximation also holds for the *safe, distributed setup* with an extra error term due to the fact that the estimated feasible sets are different across agents.

The proofs of our results are provided in the Appendix.

## 2 Related Literature

**I) Centralized Online Optimization:** For static regret, as defined in (1), it is well-known that the optimal regret bound is $O(\sqrt{T})$ (respectively, $O(\log T)$) for convex (respectively, exp-concave and strongly convex) problems (Hazan, 2016). For dynamic regret, various regularity measures have been considered. Let us first define the dynamic regret with respect to a general comparator sequence $\{\mathbf{u}_t\}_{t=1}^T$ and the corresponding regularity measure called path-length as follows

$$
\mathbf{Reg}_T^d(\mathbf{u}_1, \ldots, \mathbf{u}_T) \triangleq \sum_{t=1}^T f_t(\mathbf{x}_t) - \sum_{t=1}^T f_t(\mathbf{u}_t),
$$

$$
C_T(\mathbf{u}_1, \ldots, \mathbf{u}_T) \triangleq \sum_{t=2}^T \left\| \mathbf{u}_t - \mathbf{u}_{t-1} \right\|.
$$

(5)

The regret definition above is widely known as *universal* dynamic regret. Zinkevich (2003) showed that for the convex functions, OGD attains an upper bound of $O(\sqrt{T}(1 + C_T))$ on the universal dynamic regret (as defined in (5)). This universal dynamic regret bound was later improved to $O(\sqrt{T(1 + C_T)})$ using expert advice by Zhang et al. (2018a). More recently, for the class of comparator sequences with path length $\sum_{t=2}^T \left\| \mathbf{u}_t - \mathbf{u}_{t-1} \right\|_1 \leq C$, Baby & Wang (2021) presented, for exp-concave losses, universal dynamic regret bounds of $\tilde{O}(d^{3.5}(\max\{T^{1/3}C^{2/3}, 1\}))$, where $d$ is the problem dimension and $\tilde{O}(\cdot)$ hides the logarithmic factors when $C \geq 1/T$ and of $O(d^{1.5} \log(T))$ otherwise. Later Baby & Wang (2022) presented $\tilde{O}(\max\{d^{1/3}T^{1/3}C^{2/3}, d\})$ universal dynamic regret bound for strongly convex losses. For dynamic regret defined with respect to the minimizer sequence (as in (2)), Mokhtari et al. (2016) showed a regret bound of $O(C_T^*)$ for strongly convex and smooth functions with OGD. A related notion of higher-order path-length defined as $C_{p,T}^* \triangleq \sum_{t=2}^T \left\| \mathbf{x}_t^* - \mathbf{x}_{t-1}^* \right\|^p$ has also been considered by several works. When the minimizer sequence $\{\mathbf{x}_t^*\}_{t=1}^T$ is uniformly bounded, $O(C_{p,T}^*)$ implies $O(C_{q,T}^*)$ for $q < p$. Zhang et al. (2017a) proved that with multiple gradient queries per round, the dynamic regret is improved to $O(\min\{C_T^*, C_{2,T}^*\})$.

**II) Distributed Online Optimization:** Yan et al. (2012) studied distributed OGD for online optimization and proved a static regret bound of $O(\sqrt{T})$ (respectively, $O(\log T)$) for convex (respectively, strongly-convex) functions. Distributed online optimization for time-varying network structures was then considered in (Mateos-Núñez & Cortés, 2014; Akbari et al., 2015; Hosseini et al., 2016). Shahrampour & Jadbabaie (2018) proposed a distributed OMD algorithm with a dynamic regret bound of $O(\sqrt{T}(1 + C_T^*))$. Dixit et al. (2019a) considered time-varying network structures and showed that distributed proximal OGD achieves a dynamic regret of $O(\log T(1 + C_T^*))$ for strongly convex functions. Zhang et al. (2019) developed a method based on gradient tracking and derived a regret bound in terms of $C_T^*$ and a gradient path-length. More recently, Eshraghi & Liang (2022) showed that dynamic regret for strongly convex and smooth functions can be improved to $O(1 + C_T^*)$ using both primal and dual information boosted with multiple consensus steps. The non-convex case is also recently studied by Lu & Wang (2021), where the regret is characterized by the first-order optimality condition. On the other hand, for the projection free setup, Zhang et al. (2017b) presented a distributed online conditional gradient algorithm with a static regret bound of $O(T^{3/4})$ and a

communication complexity of $O(T)$. The communication complexity was further improved to $O(\sqrt{T})$ in (Wan et al., 2020).

Nevertheless, the works mentioned in **(I)** and **(II)** do not consider neither long-term nor safety constraints, which are discussed next.

Table 1: Related works on centralized and distributed constrained online optimization for general functions with regret and constraint violation (CV) guarantees. Let $g(\mathbf{x}) = (g_1(\mathbf{x}), g_2(\mathbf{x}), \ldots, g_n(\mathbf{x}))^\top$ be the vector formed by $n$ convex constraints. Let $c \in (0,1)$, $\alpha_0 > 1$ and $[a]_+ = \max\{0, a\}$. Problem type 'C' stands for centralized, 'D' stands for distributed (or decentralized), 'CNX' stands for convex cost functions, and 'N-CNX' stands for non-convex cost functions. Notes: † : The CV bound in (Yu & Neely, 2020) can be further reduced to $O(1)$ under a Slater's condition assumption.

| CV Type | Problem | Reference | Regret Type | Regret Bound | CV Bound |
|---|---|---|---|---|---|
| $\sum_{t=1}^T g_i(\mathbf{x}_t)\ \forall i\in[n]$ | C, CNX | (Mahdavi et al., 2012) | Static | $O(\sqrt{T})$ | $O(T^{3/4})$ |
| $\sum_{t=1}^T \max_{i\in[n]} g_i(\mathbf{x}_t)$ | C, CNX | (Jenatton et al., 2016) | Static | $O(T^{\max\{c,1-c\}})$ | $O(T^{1-c/2})$ |
| $\sum_{t=1}^T [g_i(\mathbf{x}_t)]_+\ \forall i\in[n]$ | C, CNX | (Yuan & Lamperski, 2018) | Static | $O(T^{\max\{c,1-c\}})$ | $O(T^{1-c/2})$ |
| $\sum_{t=1}^T \|[g(\mathbf{x}_t)]_+\|$ | C, CNX | (Yi et al., 2021) | Static | $O(T^{\max\{c,1-c\}})$ | $O(T^{(1-c)/2})$ |
| $\sum_{t=1}^T \|[g(\mathbf{x}_t)]_+\|$ | C, CNX | (Yi et al., 2021) | Dynamic | $O(\sqrt{T(1+C_T)})$ | $O(\sqrt{T})$ |
| $\sum_{t=1}^T g_i(\mathbf{x}_t)\ \forall i\in[n]$ | C, CNX | (Yu & Neely, 2020) | Static | $O(\sqrt{T})$ | $O(T^{1/4})†$ |
| $\left[\sum_{t=1}^T g_i(\mathbf{x}_{j,t})\right]_+, i=1, \forall j\in[m]$ | D, CNX | (Yuan et al., 2017) | Static | $O(T^{1/2+c/2})$ | $O(T^{1-c/4})$ |
| $\sum_{t=1}^T \sum_{j=1}^m \sum_{i=1}^n [\mathbf{a}_i^\top \mathbf{x}_{j,t}]_+$ | D, CNX | (Yuan et al., 2020) | Static | $O(\sqrt{T})$ | $O(T^{3/4})$ |
| $\sum_{t=1}^T \sum_{j=1}^m \sum_{i=1}^n [g_i(\mathbf{x}_{j,t})]_+$ | D, CNX | (Yuan et al., 2021) | Static | $O(T^{\max\{c,1-c\}})$ | $O(T^{1-c/2})$ |
| $\frac{1}{m}\sum_{i=1}^m \sum_{t=1}^T \left\|[g_t(\mathbf{x}_{i,t})]_+\right\|$ | D, CNX | (Yi et al., 2022) | Dynamic | $O((\alpha_0^2 T^{1-c}+T^c(1+C_T))/\alpha_0)$ | $O(\sqrt{(\alpha_0+1)T^{2-c}})$ |
| $\sum_{t=1}^T \|[\mathbf{A}\mathbf{x}_t - \mathbf{b}]_+\|$ | C, CNX | (Chaudhary & Kalathil, 2022) | Static | $O(\sqrt{\log(T)}T^{2/3})$ | $0$ |
| $\sum_{t=1}^T \sum_{j=1}^m \|[\mathbf{A}\mathbf{x}_{j,t} - \mathbf{b}]_+\|$ | D, CNX | **This work** | Dynamic | $O\left(T^{2/3}\sqrt{\log T} + T^{1/3}C_T^*\right)$ | $0$ |
| $\sum_{t=1}^T \sum_{j=1}^m \|[\mathbf{A}\mathbf{x}_{j,t} - \mathbf{b}]_+\|$ | D, N-CNX | **This work** | Dynamic | $O\left(T^{2/3}\sqrt{\log T} + T^{2/3}C_T^*\right)$ | $0$ |

**III) Constrained Online Optimization:**   Practical systems come with inherent system imposed constraints on the decision variable. Some examples of such constraints are inventory/budget constraints in one-way trading problem (Lin et al., 2019) and time coupling constraints in networked distributed energy systems (Fan et al., 2020). For a known constraint set, projecting the decisions back to the constraint set is a natural way to incorporate constraints in online convex optimization (OCO). Here, projected OGD for general cost functions achieves $O(\sqrt{T})$ static regret. However, for complex constraints, projection can induce computational burden. An early work by Hazan & Kale (2012) solves the constrained optimization problem by replacing the quadratic convex program with simpler linear program using Frank-Wolfe. For understanding the following references better, let $\mathcal{X} = \{\mathbf{x} \in \mathrm{X} \subseteq \mathrm{R}^d : g(\mathbf{x}) \leq 0\}$ where $g(\mathbf{x}) = (g_1(\mathbf{x}), g_2(\mathbf{x}), \ldots, g_n(\mathbf{x}))^\top$ is the vector formed by $n$ convex constraints, and X is a closed convex set. The work by Mahdavi et al. (2012) proposed to use a simpler closed form projection in place of true desired projection attaining $O(\sqrt{T})$ static regret with $\sum_{t=1}^T g_i(\mathbf{x}_t) \leq O(T^{3/4})$ constraint violation $\forall i \in [n]$. Thus, their method achieves optimal regret with lesser computation burden at the cost of incurring constraint violations. The follow up work by Jenatton et al. (2016) proposed adaptive step size variant of Mahdavi et al. (2012) with $O\left(T^{\max\{c,1-c\}}\right)$ static regret and $O\left(T^{1-c/2}\right)$ constraint violation for $c \in (0,1)$. These bounds were further improved in Yu & Neely (2020) with a static regret bound of $O(\sqrt{T})$ and constraint violation bound of $O(T^{1/4})$. Here, the constraint violation is further reduced to $O(1)$ when $g_i(\mathbf{x})$ satisfy Slater's condition. The work by Yuan & Lamperski (2018) considered stricter 'cumulative' constraint violations of the form $\sum_{t=1}^T [g_i(\mathbf{x}_t)]_+\ \forall i \in [n]$ and proposed algorithms with $O\left(T^{\max\{c,1-c\}}\right)$ static regret and $O\left(T^{1-c/2}\right)$ 'cumulative' constraint violation for $c \in (0,1)$. For strongly convex functions, Yuan & Lamperski (2018) proved $O(\log(T))$ static regret and the constraint violation of respective form is $O(\sqrt{\log(T)T})$. More recently, the work by Yi et al. (2021) proposed an algorithm with $O\left(T^{\max\{c,1-c\}}\right)$ regret and $\sum_{t=1}^T \left\|[g(\mathbf{x}_t)]_+\right\| \leq O\left(T^{(1-c)/2}\right)$ 'cumulative' constraint violation. For strongly convex functions, Yi et al. (2021) reduced both static regret and constraint violation bounds to $O(\log(T))$. Further, Yi et al. (2021) presented a bound of $O(\sqrt{T(1+C_T)})$ for dynamic regret with an $O(\sqrt{T})$ 'cumulative' constraint violation. The algorithms in Mahdavi et al. (2012); Jenatton et al. (2016); Yu & Neely (2020); Yuan & Lamperski (2018); Yi et al. (2021) employ some flavor of online primal-dual algorithms. A series of recent works (Sun et al., 2017; Chen et al., 2017; Neely & Yu, 2017; Yu et al., 2017; Cao & Liu, 2018; Liu et al., 2022) have also dealt with time-varying constraints. Yu et al. (2017) specifically work with 'stochastic' time varying constraints.

More recent works (Yuan et al., 2017; 2020; 2021; Yi et al., 2022) have looked at distributed OCO with long term constraints. The work by Yuan et al. (2017) proposed a consensus based primal-dual sub-gradient algorithm with $O(T^{1/2+\beta_0})$ regret and $O(T^{1-\beta_0/2})$ constraint violation for $\beta_0 \in (0, 0.5)$. Single constraint function was considered in (Yuan et al., 2017), where constraint violation is of the form $[\sum_{t=1}^{T} g_i(\mathbf{x}_{j,t})]_+$, $i = 1, \forall j \in [m]$. Yuan et al. (2020) proposed algorithms for distributed online linear regression with $O(\sqrt{T})$ regret and $O(T^{3/4})$ constraint violation. Here, constraint violation takes the form $\sum_{t=1}^{T} \sum_{j=1}^{m} \sum_{i=1}^{n} [\mathbf{a}_i^\top \mathbf{x}_{j,t}]_+$, where $\mathbf{a}_i, i \in [n]$ is a constraint vector. Another primal-dual algorithm was presented in Yuan et al. (2021) with $O(T^{\max\{1-c,c\}})$ regret and $O(T^{1-c/2})$ constraint violation of the form $\sum_{t=1}^{T} \sum_{j=1}^{m} \sum_{i=1}^{n} [g_i(\mathbf{x}_{j,t})]_+$ for $c \in (0,1)$. In all of (Yuan et al., 2017; 2020; 2021) constraint functions are known a priori. More recently, Yi et al. (2022) proposed algorithms for distributed OCO with time-varying constraints, and for stricter 'network' constraint violation metric of the form $\frac{1}{m} \sum_{i=1}^{m} \sum_{t=1}^{T} \left\| [g_t(\mathbf{x}_{i,t})]_+ \right\|$. The algorithm in (Yi et al., 2022) gives a dynamic regret of $O((\alpha_0^2 T^{1-c} + T^c(1 + C_T))/\alpha_0)$ with $O(\sqrt{(\alpha_0 + 1)T^{2-c}})$ constraint violation for $\alpha_0 > 1$ and $c \in (0,1)$. Additionally, constrained distributed OCO with coupled inequality constraints is considered in (Yi et al., 2020a;b); with bandit feedback on cost function is considered in (Li et al., 2020); with partial constraint feedback is studied in (Sharma et al., 2021). For more references in this problem space, we refer the readers to the survey in (Li et al., 2022).

**IV) Safe Online Optimization:** Safe online optimization is a fairly nascent field with only a few works studying per-time safety in optimization problems. Amani et al. (2019); Khezeli & Bitar (2020) study the problem of safe linear bandits giving $O(\log(T)\sqrt{T})$ regret with no constraint violation, albeit under an assumption that a lower bound on the distance between the optimal action and safe set's boundary is known. Without the knowledge of such a lower bound, Amani et al. (2019) show $O(\log(T)T^{2/3})$ regret. Safe convex and non-convex optimization is studied in (Usmanova et al., 2019; Fereydounian et al., 2020). Safety in the context of OCO is studied in Chaudhary & Kalathil (2022) with a regret of $O(\sqrt{\log(T)}T^{2/3})$.

**Remark 1.** *Different from the works listed above, we study the problem of safe distributed online optimization with unknown linear constraints. We consider both convex and non-convex cost functions.*

## 3  Preliminaries

### 3.1  Notation

| | | |
|---|---|---|
| $[m]$ | | The set $\{1, 2, \dots, m\}$ for any integer $m$ |
| $\|\cdot\|_F$ | | Frobenius norm of a matrix |
| $\|\mathbf{X}\|_\mathbf{V}$ | | $\sqrt{\text{trace}(\mathbf{X}^\top \mathbf{V} \mathbf{X})}$, for a matrix $\mathbf{X}$ and a positive-definite matrix $\mathbf{V}$ |
| $\Pi_\mathcal{X}[\cdot]$ | | The operator for the projection to set $\mathcal{X}$ |
| $[\mathbf{A}]_{ij}$ | | The entry in the $i$-th row and $j$-th column of $\mathbf{A}$ |
| $[\mathbf{A}]_{i,:}$ | | The $i$-th row of $\mathbf{A}$ |
| $[\mathbf{A}]_{:,j}$ | | The $j$-th column of $\mathbf{A}$ |
| $\mathbf{1}$ | | The vector of all ones |
| $\mathbf{e}_i$ | | The $i$-th basis vector |
| $J_f(\mathbf{x})$ | | The Jacobian of a mapping $f(\cdot)$ at $\mathbf{x}$ |

### 3.2  Strong Convexity and Bregman Divergence

**Definition 1.** *A function $f : \mathcal{X} \to \mathrm{R}$ is $\mu$-strongly convex ($\mu > 0$) over the convex set $\mathcal{X}$ if*

$$f(\mathbf{x}) \geq f(\mathbf{y}) + \nabla f(\mathbf{y})^\top (\mathbf{x} - \mathbf{y}) + \frac{\mu}{2} \|\mathbf{x} - \mathbf{y}\|^2, \quad \forall \mathbf{x}, \mathbf{y} \in \mathcal{X}.$$

**Definition 2.** *For a strongly convex function $\phi(\cdot)$, the Bregman divergence w.r.t. $\phi(\cdot)$ over $\mathcal{X}$ is defined as*

$$\mathcal{D}_\phi(\mathbf{x}, \mathbf{y}) \triangleq \phi(\mathbf{x}) - \phi(\mathbf{y}) - \nabla \phi(\mathbf{y})^\top (\mathbf{x} - \mathbf{y}), \quad \mathbf{x}, \mathbf{y} \in \mathcal{X}.$$

### 3.3 Network Structure

The underlying network topology is governed by a symmetric doubly stochastic matrix $\mathbf{P}$, i.e., $[\mathbf{P}]_{ij} \geq 0, \forall i, j \in [m]$, and each row (or column) is summed to one. If $[\mathbf{P}]_{ij} > 0$, agents $i$ and $j$ are considered neighbors, and agent $i$ assigns the weight $[\mathbf{P}]_{ij}$ to agent $j$ when they communicate with each other. We assume that the graph structure captured by $\mathbf{P}$ is connected, i.e., there is a (potentially multi-hop) path from any agent $i$ to another agent $j \neq i$. Each agent is considered as a neighbor of itself, i.e., $[\mathbf{P}]_{ii} > 0$ for any $i \in [m]$. These constraints on the communication protocol imply a geometric mixing bound for $\mathbf{P}$ (Liu, 2008), such that $\sum_{j=1}^{m} \left| [\mathbf{P}^k]_{ji} - 1/m \right| \leq \sqrt{m} \beta^k$, for any $i \in [m]$, where $\beta$ is the second largest singular value of $\mathbf{P}$.

**Remark 2.** *In all of the algorithms proposed in the paper, we will see $\mathbf{P}$ as an input. This does not contradict the decentralized nature of the algorithms, as agent $i$ only requires the knowledge of $[\mathbf{P}]_{ji} > 0$ for any $j$ in its neighborhood. The knowledge of $\mathbf{P}$ is not global, and each agent only has local information about it.*

## 4 Safe Set Estimation

To keep the regret small, we first need to identify the linear safety constrains. It is impossible to learn the safety constraints if the algorithm receives no information that can be used to estimate the unknown constraints (Chaudhary & Kalathil, 2022). In our problem setup, we assume that the algorithm receives noisy observations of the form

$$\hat{\mathbf{x}}_{i,t} = \mathbf{A}\mathbf{x}_{i,t} + \mathbf{w}_{i,t} \qquad \forall i \in [m],$$

at every time step $t$, where the nature of noise $\mathbf{w}_{i,t}$ is described below. Here, $\mathbf{A} \in \mathrm{R}^{n \times d}, \mathbf{b} \in \mathrm{R}^n$, and $n$ is the number of constraints. Note that all agent updates are synchronous.

### 4.1 Assumptions

We make the following assumptions common to both the convex and the non-convex problem settings.

**Assumption 1.** *The set $\mathcal{X}^s$ is a closed polytope, hence, convex and compact. Also, $\|\mathbf{x}\| \leq L, \forall \mathbf{x} \in \mathcal{X}^s$, and $\max_{i \in [n]} \left\| [\mathbf{A}]_{i,:} \right\|_2 \leq L_A$.*

**Assumption 2.** *The constraint noise sequence $\{\mathbf{w}_{i,t}, t \in [T]\}$ is $R$-sub-Gaussian with respect to the filtration $\{\mathcal{F}_{i,t}, t \in [T]\}$, i.e., $\forall t \in [T], \forall i \in [m], \mathbb{E}[\mathbf{w}_{i,t}|\mathcal{F}_{i,t-1}] = 0$ and we have for any $\sigma \in \mathrm{R}$*

$$\mathbb{E}[\exp(\sigma \mathbf{x}^\top \mathbf{w}_{i,t}) \,|\, \mathcal{F}_{t-1}] \leq \exp(\sigma^2 R^2 \|\mathbf{x}\|^2 / 2).$$

**Assumption 3.** *Every agent has knowledge of a safe baseline action $\mathbf{x}^s \in \mathcal{X}^s$ such that $\mathbf{A}\mathbf{x}^s = \mathbf{b}^s < \mathbf{b}$. The agents are aware of $\mathbf{x}^s$ and $\mathbf{b}^s$ and thus, the safety gap $\Delta^s \triangleq \min_{i \in [n]}(b_i - b_i^s)$, where $b_i$ (respectively, $b_i^s$) denotes the $i$-th element of $\mathbf{b}$ (respectively, $\mathbf{b}^s$).*

The first assumption is typical to online optimization, and the second assumption on the noise is standard. The third assumption stems from the requirement to be absolutely safe at every time step. The assumption warrants the need for a safe starting point which is readily available in most practical problems of interest. Similar assumptions can be found in previous literature on safe linear bandits (Amani et al., 2019; Khezeli & Bitar, 2020), safe convex and non-convex optimization (Usmanova et al., 2019; Fereydounian et al., 2020), and safe online convex optimization (Chaudhary & Kalathil, 2022).

### 4.2 Explore and Estimate

In this section, we present an algorithmic subroutine, Algorithm 1, for agents to obtain sufficiently good local estimates of $\mathcal{X}^s$ before beginning to perform OGD. For the first $T_0$ time steps, each agent safely explores around the baseline action $\mathbf{x}^s$. Each exploratory action is a $\gamma$-weighted combination of the baseline action and an i.i.d random vector $\zeta_{i,t}$. Here, for the agent $i \in [m]$ at time step $t \in [T_0], \gamma \in [0, 1)$, and $\zeta_{i,t}$ is zero mean i.i.d random vector with $\|\zeta_{i,t}\| \leq L$ and $Cov(\zeta_{i,t}) = \sigma_\zeta^2 \mathbf{I}$. Performing exploration in this manner ensures per time step safety requirement as noted in Lemma 1. The proof of lemma is immediate from the assumptions.

**Lemma 1.** *(Lemma 1 in Chaudhary & Kalathil (2022)) Let Assumptions 1-3 hold. With $\gamma = \frac{\Delta^s}{LL_A}$, $\mathbf{A}\mathbf{x}_{i,t} \leq \mathbf{b}$ for each $\mathbf{x}_{i,t} = (1-\gamma)\mathbf{x}^s + \gamma\zeta_{i,t} \ \forall i \in [m], t \in [T_0]$.*

Once the data collection phase is finished, each agent $i \in [m]$ constructs local function $l_i(\mathbf{A})$ of the form

$$l_i(\mathbf{A}) \triangleq \sum_{t=1}^{T_0} \left\| \mathbf{A}\mathbf{x}_{i,t} - \hat{\mathbf{x}}_{i,t} \right\|^2 + \frac{\lambda}{m} \left\| \mathbf{A} \right\|_F^2.$$

Then, for time steps $t \in [T_0 + 1, T_0 + T_1]$, Alg. **EXTRA** (Shi et al., 2015) is used to solve the global Least Squares (LS) estimation problem $\sum_{i=1}^m l_i(\mathbf{A})$ in a distributed fashion with a proper choice of $\alpha$.

**Lemma 2.** *Suppose Assumptions 1-2 hold. Let Algorithm 1 run with $T_0 = \Omega(\frac{L^2}{m\gamma^2\sigma_\zeta^2}\log(\frac{d}{\delta}))$ for data collection and $T_1 = \Theta(\log T^\rho)$, where $\rho$ is a positive constant. Denote the final output of the algorithm as $\widehat{\mathbf{A}}_i$ for agent $i \in [m]$. Then, with probability at least $(1 - 2\delta)$, we have $\forall k \in [n]$ and $\forall i, j \in [m]$*

$$\left\| [\widehat{\mathbf{A}}_i]_{k,:} - [\mathbf{A}]_{k,:} \right\| \leq \frac{1}{T^\rho} + \frac{R\sqrt{d\log\left(\frac{1+mT_0L^2/\lambda}{\delta/n}\right)} + \sqrt{\lambda}L_A}{\sqrt{\frac{1}{2}m\gamma^2\sigma_\zeta^2 T_0}}, \tag{6}$$

$$\left\| [\widehat{\mathbf{A}}_i]_{k,:} - [\widehat{\mathbf{A}}_j]_{k,:} \right\| \leq \frac{2}{T^\rho},$$

*where $[\widehat{\mathbf{A}}_i]_{k,:}$ and $[\mathbf{A}]_{k,:}$ are the k-th rows of $\widehat{\mathbf{A}}_i$ and $\mathbf{A}$, respectively.*

It is worth noting that the safety gap $\Delta^s$ affects the estimation error according to Lemma 2. As we mentioned earlier, the exploratory action $\mathbf{x}_{i,t} = (1-\gamma)\mathbf{x}^s + \gamma\zeta_{i,t}$, where the coefficient $\gamma = \frac{\Delta^s}{LL_A}$. Intuitively, if $\Delta^s$ is larger, we can put more weight on the exploration through $\zeta_{i,t}$, which is beneficial to the estimation accuracy. We see from Equation (6) that when $\gamma$ is larger, the estimation error bound is tighter.

Let us also discuss the computational complexity of Algorithm 1. The time cost of the data-collection phase is $O(mdT_0)$, assuming that it takes $O(d)$ time to compute each action. For the estimation phase, to perform a single update, each agent spends $O(mnd)$ time for the calculation of the weighted average and $O(T_0nd)$ time for the gradient computation. Accordingly, the total time cost of Algorithm 1 is $O\big(mT_1(mnd + T_0nd)\big)$.

Let us now define the estimated safe set for each agent $i \in [m]$. Let the parameter estimate for agent $i \in [m]$ at the end of $T_0 + T_1$ time step be denoted by $\widehat{\mathbf{A}}_i$. For each row $k \in [n]$ of $\widehat{\mathbf{A}}_i$, a ball centered at $[\widehat{\mathbf{A}}_i]_{k,:}$ with a radius of $\mathcal{B}_r$ can be defined as follows

$$\mathcal{C}_{i,k} \triangleq \{\mathbf{a} \in \mathrm{R}^d : \left\| \mathbf{a} - [\widehat{\mathbf{A}}_i]_{k,:} \right\| \leq \mathcal{B}_r\}, \tag{7}$$

where $\mathcal{B}_r \triangleq \frac{1}{T^\rho} + \frac{R\sqrt{d\log\left(\frac{1+mT_0L^2/\lambda}{\delta/n}\right)} + \sqrt{\lambda}L_A}{\sqrt{\frac{1}{2}m\gamma^2\sigma_\zeta^2 T_0}}$. The true parameter $[\mathbf{A}]_{k,:}$ lies inside the set $\mathcal{C}_{i,k}$ with a high probability. Now, using (7) the safe estimated set for agent $i \in [m]$ can be constructed as follows:

$$\widehat{\mathcal{X}}_i^s \triangleq \{\mathbf{x} \in \mathrm{R}^d : \tilde{\mathbf{a}}_k^\top \mathbf{x} \leq b_k, \ \forall \tilde{\mathbf{a}}_k \in \mathcal{C}_{i,k}, \ \forall k \in [n]\}. \tag{8}$$

The safe estimated set above will be used by each agent for the projection step in subsequent algorithms.

## 5 Dynamic Regret Analysis for the Convex Setting

During the first $T_0 + T_1$ time steps in Algorithm 1 agents do not expend any effort to minimize the regret. This is due to the fact that without the knowledge of the feasible set, they cannot perform any projection. In this section, we propose D-Safe-OGD, which allows agents to carry out a safe distributed online optimization, and we analyze D-Safe-OGD in the *convex* setting.

D-Safe-OGD is summarized in Algorithm 2, where in the exploration phase, all agents collaboratively estimate the constraint parameters based on Algorithm 1, and then each agent constructs the feasible set based on its own estimate.

---

**Algorithm 1** Distributed Constraint Parameter Estimation

---

1: **Require:** number of agents $m$, doubly stochastic matrix $\mathbf{P} \in \mathrm{R}^{m \times m}$, $\tilde{\mathbf{P}} \triangleq \frac{\mathbf{I}+\mathbf{P}}{2}$, hyper-parameters $\alpha$, $\gamma$ and $\lambda$, data-collection duration $T_0$, constraint-estimation duration $T_1$, a strictly feasible point $\mathbf{x}^s$ (safe baseline action).
2: **Explore around baseline action:**
3: **for** $t = 1, 2, \ldots, T_0$ **do**
4:    **for** $i = 1, 2, \ldots, m$ **do**
5:       Select action $\mathbf{x}_{i,t} = (1 - \gamma)\mathbf{x}^s + \gamma\zeta_{i,t}$
6:       Receive noisy observation $\hat{\mathbf{x}}_{i,t} = \mathbf{A}\mathbf{x}_{i,t} + \mathbf{w}_{i,t}$
7:    **end for**
8: **end for**
9: **Form local functions using collected data:**

$$l_i(\mathbf{A}) \triangleq \sum_{t=1}^{T_0} \left\| \mathbf{A}\mathbf{x}_{i,t} - \hat{\mathbf{x}}_{i,t} \right\|^2 + \frac{\lambda}{m} \left\| \mathbf{A} \right\|_F^2.$$

10: **Use EXTRA** (Shi et al., 2015) **to solve global LS problem** $\sum_{i=1}^m l_i(\mathbf{A})$ **in a distributed fashion:**
11: Randomly generate $\widehat{\mathbf{A}}_i^{T_0} \in \mathrm{R}^{n \times d}$ for all $i \in [m]$.
12: $\forall i \in [m]$, $\widehat{\mathbf{A}}_i^{T_0+1} = \sum_{j=1}^m [\mathbf{P}]_{ji} \widehat{\mathbf{A}}_j^{T_0} - \alpha \nabla l_i(\widehat{\mathbf{A}}_i^{T_0})$, where the gradient is computed based on the following expression:

$$\nabla l_i(\mathbf{A}) = \sum_{t=1}^{T_0} \left[ 2\mathbf{A}\mathbf{x}_{i,t}\mathbf{x}_{i,t}^\top - 2\hat{\mathbf{x}}_{i,t}\mathbf{x}_{i,t}^\top \right] + \frac{2\lambda}{m}\mathbf{A}.$$

13: **for** $t = T_0, \ldots, T_0 + T_1 - 2$ **do**
14:    **for** $i = 1, 2, \ldots, m$ **do**
15:       $\widehat{\mathbf{A}}_i^{t+2} = \sum_{j=1}^m 2[\tilde{\mathbf{P}}]_{ji} \widehat{\mathbf{A}}_j^{t+1} - \sum_{j=1}^m [\tilde{\mathbf{P}}]_{ji} \widehat{\mathbf{A}}_j^t - \alpha[\nabla l_i(\widehat{\mathbf{A}}_i^{t+1}) - \nabla l_i(\widehat{\mathbf{A}}_i^t)]$.
16:    **end for**
17: **end for**

---

In the optimization phase, the network applies distributed OGD, where all agents first perform gradient descent with their local gradients, and then they communicate their iterates with neighbors based on the network topology imposed by $\mathbf{P}$. We note that the projection operator of each agent is defined w.r.t. the local estimated feasible set (line 8 of Algorithm 2), thereby making the feasible sets close enough but slightly different from each other. Therefore, previous regret bounds for distributed online optimization over a common feasible set (e.g., (Yan et al., 2012; Hosseini et al., 2016; Shahrampour & Jadbabaie, 2018; Eshraghi & Liang, 2022)) are not immediately applicable. We tackle this challenge by exploiting the geometric property of linear constraints (Fereydounian et al., 2020) as well as the sensitivity analysis of perturbed optimization problems with second order regular constraints (Bonnans et al., 1998), and we present an upper bound on the dynamic regret in terms of the path-length regularity measure.

We adhere to the following standard assumption in the context of OCO:

**Assumption 4.** *The cost functions $f_{i,t}$ are convex $\forall i \in [m]$ and $\forall t \in [T]$, and they have a bounded gradient, i.e., $\left\| \nabla f_{i,t}(\mathbf{x}) \right\| \leq G$ for any $\mathbf{x} \in \mathcal{X}^s$.*

**Theorem 3.** *Suppose Assumptions 1-4 hold and $T = \Omega\left( \left(\frac{L^2}{m\gamma^2\sigma_\zeta^2} \log(\frac{d}{\delta})\right)^{3/2} \right)$. By running Algorithm 2 with $\gamma \leq \frac{\Delta^s}{LL_A}$, $\eta = \Theta(T^{-1/3})$, $T_0 = \Theta(T^{2/3})$ and $T_1 = \Theta(\log T)$, we have with probability at least $(1 - 2\delta)$*

$$\mathbf{x}_{i,t} \in \mathcal{X}^s, \ \forall i \in [m], t \in [T], \quad and$$

$$\mathbf{Reg}_{i,T}^d = O\left( T^{2/3}\sqrt{\log(T/\delta)} + \frac{\beta}{(1-\beta)}T^{2/3} + T^{1/3}C_T^* \right), \ \forall i \in [m].$$

Theorem 3 establishes a dynamic regret bound for D-Safe-OGD that is at least $O(T^{2/3}\sqrt{\log T})$, and for a large enough path-length the bound becomes $O(T^{1/3}C_T^*)$. We can also see the impact of network topology through $\beta$, the second

---

**Algorithm 2** Distributed Safe OGD with linear constraints

---

1: **Require:** number of agents $m$, doubly stochastic matrix $\mathbf{P} \in \mathrm{R}^{m \times m}$, hyper-parameters $\alpha, \gamma, \eta, \delta, \lambda$, time horizon $T$, a strictly feasible point $\mathbf{x}^s$.

2: Specify $T_0$ and $T_1$ based on given hyper-parameters and run Algorithm 1 to learn agents estimates $\{\hat{\mathbf{A}}_i\}_{i \in [m]}$ in a distributed fashion.

3: For all $i \in [m]$, construct the safe set $\widehat{\mathcal{X}}_i^s$ from the estimate $\hat{\mathbf{A}}_i$.

4: **Distributed online gradient descent over different feasible sets:**

5: Let $T_s \triangleq (T_0 + T_1 + 1)$ and initialize all agents at the same point $\mathbf{x}_{i,T_s} = \mathbf{x}_{T_s}$ chosen randomly.

6: **for** $t = T_s, \ldots, T$ **do**

7:   **for** $i = 1, 2, \ldots, m$ **do**

8:
$$\mathbf{y}_{i,t} = \Pi_{\widehat{\mathcal{X}}_i^s} \left[ \mathbf{x}_{i,t} - \eta \nabla f_{i,t}(\mathbf{x}_{i,t}) \right].$$

9:   **end for**

10:   For all $i \in [m]$,

11:
$$\mathbf{x}_{i,t+1} = \sum_{j=1}^{m} [\mathbf{P}]_{ji} \mathbf{y}_{j,t}.$$

12: **end for**

---

largest singular value of $\mathbf{P}$. When the network connectivity is stronger (i.e., $\beta$ is smaller), the regret bound is tighter. For the dependence on other parameters, we refer readers to the exact upper bound expression (Equation (38)).

**Corollary 4.** *Suppose that the comparator sequence is fixed over time, i.e., $\mathbf{x}_t^* = \mathbf{x}^*$, $\forall t \in [T]$. Then, the individual regret bound is $O(T^{2/3}\sqrt{\log T})$, which recovers the static regret bound of the centralized case in (Chaudhary & Kalathil, 2022) in terms of order.*

**Remark 3.** *Note that when $\mathbf{A}$ is known, there is no estimation error, and the trade-off in terms of $\eta$ and $T_0$ no longer exists. In other words, the agents do not incur the initial regret of $T_0 + T_1 = O(T^{2/3})$, caused by estimation. Then, by choosing $\eta = \Theta(\frac{1}{\sqrt{T}})$, the resulting bound is $O(\sqrt{T}(1 + C_T^*))$, which recovers the result of Shahrampour & Jadbabaie (2018) in terms of order.*

**Remark 4.** *In the proof of Theorem 3, the regret bound is shown to be $O(T_0 + \frac{1}{\eta} + \frac{1}{\eta}C_T^* + \frac{T\sqrt{\log T_0}}{\sqrt{T_0}} + \frac{\beta \eta T}{(1-\beta)})$. If agents have the knowledge of $C_T^*$, by setting $\eta = \Theta(T^{-1/2}\sqrt{C_T^*})$, the regret bound in Theorem 3 is improved to $O(T^{2/3}\sqrt{\log T} + \sqrt{T C_T^*})$, which enjoys better dependence on $C_T^*$. Though this choice of step size is non-adaptive, we conjecture that using techniques such as expert advice (Zhang et al. (2018a)) or adaptive step sizes (Jadbabaie et al. (2015)), one can improve the regret bound, which is an interesting future direction.*

## 6   Dynamic Regret Analysis for the Non-convex Setting

In this section, we study the *non-convex* setting for safe distributed online optimization. Even for offline optimization in the non-convex setting, the standard metric for the convergence analysis is often stationarity, i.e., characterizing the decay rate of the gradient norm. In online optimization, we can also expect that standard regret notions, used in the convex setting, may not be tractable for understanding the algorithm performance. However, in a recent work by Ghai et al. (2022), the authors studied an algorithmic equivalence property between OGD and OMD for certain problem geometries, in the sense that OGD applied to non-convex problems can be approximated by OMD applied to convex functions under reparameterization, using which a sub-linear static regret bound is guaranteed.

More specifically, for a centralized problem, suppose that there is a bijective non-linear mapping $q$, such that $\mathbf{u}_t = q(\mathbf{x}_t)$, and consider the OGD and OMD updates
**Centralized OGD:**
$$\mathbf{x}_{t+1} = \operatorname{argmin}_{\mathbf{x} \in \mathcal{X}} \left\{ \nabla f_t(\mathbf{x}_t)^\top (\mathbf{x} - \mathbf{x}_t) + \frac{1}{2\eta} \|\mathbf{x} - \mathbf{x}_t\|^2 \right\}, \tag{9}$$

**Centralized OMD:**

$$\mathbf{u}_{t+1} = \text{argmin}_{\mathbf{u} \in \mathcal{X}'} \left\{ \nabla \tilde{f}_t(\mathbf{u}_t)^\top (\mathbf{u} - \mathbf{u}_t) + \tfrac{1}{\eta} \mathcal{D}_\phi(\mathbf{u}, \mathbf{u}_t) \right\}, \tag{10}$$

where $\mathcal{X}'$ is the image of $\mathcal{X}$ under the mapping $q$. Ghai et al. (2022) quantified the deviation $\left\| \mathbf{u}_{t+1} - q(\mathbf{x}_{t+1}) \right\|$ as $O(\eta^{3/2})$ under the following technical assumptions (together with boundedness of gradient norms):

**Assumption 5.** *There exists a bijective mapping $q : \mathcal{X} \to \mathcal{X}'$ such that $[\nabla^2 \phi(\mathbf{u})]^{-1} = J_q(\mathbf{x}) J_q(\mathbf{x})^\top$ where $\mathbf{u} = q(\mathbf{x})$.*

**Assumption 6.** *Let $W > 1$ be a constant. Assume that $q(\cdot)$ is $W$-Lipschitz, $\phi(\cdot)$ is $1$-strongly convex and smooth with its first and third derivatives upper bounded by $W$. The first and second derivatives of $q^{-1}(\cdot)$ are also bounded by $W$. For all $\mathbf{u} \in \mathcal{X}'$, $\mathcal{D}_\phi(\mathbf{u}, \cdot)$ is $W$-Lipschitz over $\mathcal{X}'$.*

Examples that satisfy these assumptions are provided in Section 3.1 of Ghai et al. (2022). For example, if $\phi$ is the negative entropy (respectively, log barrier), we can use quadratic (respectively, exponential) reparameterization for $q$. Amid & Warmuth (2020) showed that in the continuous-time setup when Assumption 5 holds, the mirror descent regularization induced by $\phi$ can be transformed back to the Euclidean regularization by $q^{-1}$, which implies the equivalence between OMD for convex functions and OGD for non-convex functions. This is due to the fact that higher than second order factors vanish in continuous time, and this assures that the mirror flow and the reparameterized gradient flow coincide. Though in the discrete-time case, the exact equivalence does not hold, Ghai et al. (2022) showed that OGD for non-convex functions can still be approximated as OMD for convex functions, and the corresponding static regret bound is $O(T^{2/3})$ under the assumption that $f_t(\mathbf{x}) = \tilde{f}_t(q(\mathbf{x}))$, where $\tilde{f}_t(\cdot)$ is convex. However, we need more technical assumptions to handle the discrete-time setup as higher order terms are relevant and must be judiciously analyzed. Ghai et al. (2022) characterized the sufficient condition to achieve Assumption 5, which entails an implicit OMD reparameterization for a non-convex OGD. We state these (two assumptions) by tailoring them to our problem setting:

**Assumption 7.** $\left\| \nabla \tilde{f}_{i,t}(\mathbf{u}) \right\| \leq G_F$ *for all $\mathbf{u} \in \mathcal{X}^{s\prime}$ and $\sup_{\mathbf{u}, \mathbf{z} \in \mathcal{X}^{s\prime}} \mathcal{D}_\phi(\mathbf{u}, \mathbf{z}) \leq D'$.*

**Assumption 8.** *Properties of the mapping $q(\cdot)$:*

- *There exists a mapping $q(\cdot)$ such that $f_{i,t}(\mathbf{x}) = \tilde{f}_{i,t}(q(\mathbf{x}))$, where $\tilde{f}_{i,t}(\cdot)$ is convex.*

- *$q(\cdot)$ is a $C^3$-diffeomorphism, and $J_q(\mathbf{x})$ is diagonal.*

- *For any $\mathcal{X} \subset \mathcal{X}^s$ which is compact and convex, $\mathcal{X}' \triangleq q(\mathcal{X})$ is convex and compact.*

We again refer the reader to Section 3.1 of Ghai et al. (2022) for examples related to Assumption 8.

In this work, we extend this equivalence to "distributed" variants of OGD and OMD under the additional complexity that the constraint set is unknown, and it can only be approximated via Algorithm 1. Our focus is on analyzing the effect of (i) the constraint estimation as well as (ii) the distributed setup in non-convex online learning, and we also generalize the analysis of Ghai et al. (2022) to the *dynamic* regret. For the technical analysis of the non-convex setting, we also use the following assumption.

**Assumption 9.** *Let $\mathbf{u}$ and $\{\mathbf{y}_i\}_{i=1}^m$ be vectors in $\mathrm{R}^d$. The Bregman divergence satisfies the separate convexity in the following sense*

$$\mathcal{D}_\phi\left(\mathbf{u}, \sum_i^m \alpha_i \mathbf{y}_i\right) \leq \sum_i^m \alpha_i \mathcal{D}_\phi(\mathbf{u}, \mathbf{y}_i),$$

*where $\alpha \in \Delta_m$ is on the $m-$dimensional simplex.*

This assumption is satisfied by commonly used Bregman divergences, e.g., Euclidean distance and KL divergence. We refer interested readers to (Bauschke & Borwein, 2001; Shahrampour & Jadbabaie, 2018) for more information.

In the following theorem, we prove that with high probability, the dynamic regret bound of D-Safe-OGD is $O(T^{2/3}\sqrt{\log T} + T^{2/3} C_T^*)$.

**Theorem 5.** *Suppose Assumptions 1-3 and 6-9 hold and $T = \Omega\left( \left( \frac{L^2}{m \gamma^2 \sigma_\zeta^2} \log(\frac{d}{\delta}) \right)^{3/2} \right)$. By running Algorithm 2 with $\gamma \leq \frac{\Delta^s}{L L_A}$, $\eta = \Theta(T^{-2/3})$, $T_0 = \Theta(T^{2/3})$ and $T_1 = \Theta(\log T)$, we have with probability at least $(1 - 2\delta)$*

$$\mathbf{x}_{i,t} \in \mathcal{X}^s, \ \forall i \in [m], t \in [T], \quad and$$

$$\mathbf{Reg}_{i,T}^d = O(T^{2/3}\sqrt{\log(T/\delta)} + T^{2/3}C_T^*), \ \forall i \in [m].$$

The complete proof is provided in the Appendix, and the dependence on other problem parameters can be found in Equation (54). The idea is to show that distributed OMD and distributed OGD iterates are close enough to each other if the reference points of both updates are identical, i.e., $\mathbf{u}_{i,t} = q(\mathbf{x}_{i,t})$ for all $i \in [m]$. Then, distributed OGD can be viewed as a perturbed version of distributed OMD, and under the assumption of convexity of $\tilde{f}_{i,t}$ the regret bound can be established. Also, as mentioned in the convex case (Remark 4), we conjecture that the dependence of regret bound to the path-length can be improved to $\sqrt{C_T^*}$.

We further have the following corollary that shows our result is a valid generalization of Ghai et al. (2022) to the *distributed*, *dynamic* setting.

**Corollary 6.** *Suppose that the comparator sequence is static over time, i.e., $\mathbf{x}_t^* = \mathbf{x}^*, \ \forall t \in [T]$. Then, the individual regret bound becomes $O(T^{2/3}\sqrt{\log T})$, which recovers the static regret bound of Ghai et al. (2022) up to log factors.*

It is worth noting that though in the convex case, the estimation of unknown constraints exacerbates the regret bound (due to $O(T^{2/3})$ time spent on exploration), for the non-convex case, the resulting bound still matches the static regret of Ghai et al. (2022), where there is no estimation error. In other words, there is no trade-off in this case as the static regret (without estimation error) is $O(T^{2/3})$ (Ghai et al., 2022) (disregarding log factors).

## 7  Discussion on Regret Bounds in Terms of Other Regularity Measures

In this work, we focused on dynamic regret bounds characterized by $C_T^*$, the path length of the minimizer sequence. It is worth noting that existing works also presented regret bounds in terms of other regularity measures, which capture the properties of the function sequence from different perspectives. Such measures include (1) the function variation $V_T \triangleq \sum_{t=2}^T \sup_{\mathbf{x} \in \mathcal{X}} |f_t(\mathbf{x}) - f_{t-1}(\mathbf{x})|$ (Besbes et al., 2015), (2) the predictive path-length $C_T'(\mathbf{u}_1, \ldots, \mathbf{u}_T) \triangleq \sum_{t=2}^T \|\mathbf{u}_t - \Psi_t(\mathbf{u}_{t-1})\|$ (Hall & Willett, 2013), where $\Psi_t$ is a given dynamics, and (3) the gradient variation $D_T \triangleq \sum_{t=1}^T \|\nabla f_t(\mathbf{x}_t) - \mathbf{m}_t\|^2$ (Rakhlin & Sridharan, 2013), where $\mathbf{m}_t$ is a predictable sequence computed by the learner. Besbes et al. (2015) proposed a restarting OGD algorithm and showed that when the learner has access to only noisy gradients, the expected dynamic regret is bounded by $O(T^{2/3}(V_T + 1)^{1/3})$ for convex functions and $O(\log T \sqrt{T(1 + V_T)})$ for strongly convex functions. The above mentioned regularity measures are not directly comparable to each other. In this regard, Jadbabaie et al. (2015) provided a dynamic regret bound in terms of $C_T^*$, $D_T$ and $V_T$ for the adaptive optimistic OMD algorithm. Also, Chang & Shahrampour (2021) revisited OGD with multiple gradient queries per iteration (in the unconstrained setup) and established the regret bound of $O(\min\{V_T, C_T^*, C_{2,T}^*\})$ for strongly convex and smooth functions. Dynamic regret has also been studied for functions with a parameterizable structure (Ravier et al., 2019) as well as composite convex functions (Ajalloeian et al., 2020).

Besides the dynamic regret, a relevant regret measure called *adaptive* regret (Hazan & Seshadhri, 2009) for a contiguous time interval $T_{sub}$ is defined as follows

$$\mathbf{Reg}_T^a(T_{sub}) \triangleq \max_{[i,i+T_{sub}-1] \subset [T]} \left( \sum_{t=i}^{i+T_{sub}-1} f_t(\mathbf{x}_t) - \min_{\mathbf{x} \in \mathcal{X}} \sum_{t=i}^{i+T_{sub}-1} f_t(\mathbf{x}) \right). \tag{11}$$

Zhang et al. (2018b) analyzed the connection between adaptive regret and dynamic regret and provided adaptive algorithms with provably small dynamic regret for convex, exponentially concave, and strongly convex functions.

In contrast to aforementioned works, where a projection operator is needed, Wan et al. (2021) proposed a projection free online method replacing the projection step with multiple linear optimization steps. Without assuming smoothness, they proved dynamic regret bounds of $O(\max\{T^{2/3}V_T^{1/3}, \sqrt{T}\})$ and $O(\max\{\sqrt{TV_T \log T}, \log T\})$ for convex and strongly convex functions, respectively. On the other hand, Wan et al. (2023) considered the case of smooth convex functions and improved the dynamic regret bound from $O(\sqrt{T}(1 + V_T + \sqrt{D_T}))$ to $O(\sqrt{T(1 + V_T)})$.

## Conclusion

In this work, we considered safe distributed online optimization with an unknown set of linear constraints. The goal of the network is to ensure that the action sequence selected by each agent, which only has partial information

about the global function, is competitive to the centralized minimizers in hindsight without violating the safety constraints. To address this problem, we proposed D-Safe-OGD, where starting from a safe region, it allows all agents to perform exploration to estimate the unknown constraints in a distributed fashion. Then, distributed OGD is applied over the feasible sets formed by agents estimates. For convex functions, we proved a dynamic regret bound of $O(T^{2/3}\sqrt{\log T} + T^{1/3}C_T^*)$, which recovers the static regret bound of Chaudhary & Kalathil (2022) for the centralized case (single agent). Then, we showed that for the non-convex setting, the dynamic regret is upper bounded by $O(T^{2/3}\sqrt{\log T} + T^{2/3}C_T^*)$, which recovers the static regret bound of Ghai et al. (2022) for the centralized case (single agent) up to log factors. Possible future directions include improving the regret using adaptive techniques and/or deriving comprehensive regret bounds in terms of other variation measures, such as $V_T$.

## Acknowledgements

The authors gratefully acknowledge the support of National Science Foundation (NSF). T-J. Chang and S. Shahrampour were supported by NSF ECCS-2136206. The work of S. Chaudhary and D. Kalathil was supported in part by grants NSF CAREER-EPCN-2045783 and NSF CNS-1955696.

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

# A  Appendix

In this section, we provide the proofs of our theoretical results. In Section A.1, we state the results we use in our analysis. Section A.2 includes the proof of estimation error bound in Lemma 2. In Sections A.3 and A.4, we provide the proofs for Theorem 3 (convex case) and Theorem 5 (non-convex case), respectively.

## A.1  Preliminaries:

**Theorem 7.** *(Theorem 2 in Abbasi-Yadkori et al. (2011)). Let $\{\mathcal{F}_t\}_{t=0}^{\infty}$ be a filtration and $\{w_t\}_{t=1}^{\infty}$ be a real-valued stochastic process. Here, $w_t$ is $\mathcal{F}_t$-measurable and $w_t$ is conditionally $R$-sub Gaussian for some $R \geq 0$. Let $\{\mathbf{x}_t\}_{t=1}^{\infty}$ be an $\mathrm{R}^d$-valued stochastic process such that $\mathbf{x}_t$ is $\mathcal{F}_{t-1}$-measurable. Let $\mathbf{V}_T \triangleq \sum_{t=1}^{T} \mathbf{x}_t \mathbf{x}_t^{\top} + \lambda \mathbf{I}$ where $\lambda > 0$. Define $y_t = \mathbf{a}^{\top} \mathbf{x}_t + w_t$, then $\hat{\mathbf{a}}_T = \mathbf{V}_T^{-1} \sum_{t=1}^{T} y_t \mathbf{x}_t$ is the $\ell_2$-regularized least squares estimate of $\mathbf{a}$. Assume $\|\mathbf{a}\| \leq L_A$ and $\|\mathbf{x}_t\| \leq L$, $\forall t$. Then, for any $\delta \in (0, 1)$, with probability $(1 - \delta)$, the true parameter $\mathbf{a}$ lies in the following set:*

$$\left\{ \mathbf{a} \in \mathrm{R}^d : \left\| \mathbf{a} - \hat{\mathbf{a}}_T \right\|_{\mathbf{V}_T} \leq R \sqrt{d \log \left( \frac{1 + TL^2/\lambda}{\delta} \right)} + \sqrt{\lambda} L_A \right\},$$

*for all $T \geq 1$.*

**Theorem 8.** *(Theorem 5.1.1 in Tropp et al. (2015)). Consider a finite sequence $\{\mathbf{X}_t\}$ of independent, random and positive semi-definite matrices of dimension $d$. Assume that $\lambda_{\max}(\mathbf{X}_t) \leq L$, $\forall t$. Define $\mathbf{Y} \triangleq \sum_t \mathbf{X}_t$ and denote $\lambda_{\min}(\mathbb{E}[\mathbf{Y}])$ as $\mu$. Then, we have*

$$\mathbb{P}(\lambda_{\min}(\mathbf{Y}) \leq \epsilon \mu) \leq d \exp\left( -(1 - \epsilon)^2 \frac{\mu}{2L} \right), \text{ for any } \epsilon \in (0, 1).$$

Now, let us define the *shrunk* version of the polytope as follows

$$\mathcal{X}_{\text{in}}^s \triangleq \{\mathbf{x} \in \mathrm{R}^d : [\mathbf{A}]_{k,:} \mathbf{x} + \tau_{\text{in}} \leq b_k, \forall k \in [n]\}, \text{ for some } \tau_{\text{in}} > 0. \tag{12}$$

**Lemma 9** (Lemma 1 in Fereydounian et al. (2020)). *Consider a positive constant $\tau_{\text{in}}$ such that $\mathcal{X}_{\text{in}}^s$ is non-empty. Then, for any $\mathbf{x} \in \mathcal{X}^s$,*

$$\left\| \Pi_{\mathcal{X}_{\text{in}}^s}(\mathbf{x}) - \mathbf{x} \right\| \leq \frac{\sqrt{d} \tau_{\text{in}}}{C(\mathbf{A}, \mathbf{b})}, \tag{13}$$

*where $C(\mathbf{A}, \mathbf{b})$ is a positive constant that depends only on the matrix $\mathbf{A}$ and the vector $\mathbf{b}$.*

**Theorem 10.** *(Theorem 3.7 in Shi et al. (2015)) Let us consider the following notation for EXTRA algorithm*

$$\mathbf{x}_{i,k} : \textit{The iterate of agent } i \textit{ at time } k \textit{ of the EXTRA algorithm,}$$

$$\mathbf{X}_k = \begin{bmatrix} \mathbf{x}_{1,k}^{\top} \\ \vdots \\ \mathbf{x}_{m,k}^{\top} \end{bmatrix},$$

$$\mathbf{x}^* = argmin_{\mathbf{x}} \left\{ \sum_{i=1}^{m} f_i(\mathbf{x}) \right\},$$

$$\mathbf{X}^* = \begin{bmatrix} \mathbf{x}^{*\top} \\ \vdots \\ \mathbf{x}^{*\top} \end{bmatrix},$$

$$\mathbf{f}(\mathbf{X}) = \sum_{i=1}^{m} f_i(\mathbf{x}_i).$$

*A convex function $h(\cdot)$ is restricted strongly convex w.r.t. a point $\mathbf{y}$ if there exists $\mu > 0$ such that*

$$\langle \nabla h(\mathbf{x}) - \nabla h(\mathbf{y}), \mathbf{x} - \mathbf{y} \rangle \geq \mu \|\mathbf{x} - \mathbf{y}\|^2, \forall \mathbf{x}.$$

*Suppose that the gradient of $\mathbf{f}(\mathbf{X})$ w.r.t. $\mathbf{X}$ is Lipschitz continuous with a constant $L_f$ and $\mathbf{f}(\mathbf{X}) + \frac{1}{4\alpha}\left\|\mathbf{X} - \mathbf{X}^*\right\|_{\tilde{\mathbf{P}}-\mathbf{P}}$ is restricted strongly convex w.r.t. $\mathbf{X}^*$ with a constant $\mu_g$. Then, with a proper step size $\alpha < \frac{2\mu_g\lambda_{min}(\tilde{\mathbf{P}})}{L_f^2}$, there exists $\varsigma > 0$ such that $\left\|\mathbf{X}_k - \mathbf{X}^*\right\|_{\tilde{\mathbf{P}}}^2$ converges to 0 at the R-linear rate of $O((1+\varsigma)^{-k})$.*

## A.2 Safe Distributed Set Estimation

*Proof of Lemma 2.* Let $\mathbf{V}_{T_0} \triangleq \sum_{i=1}^{m}\sum_{t=1}^{T_0} \mathbf{x}_{i,t}\mathbf{x}_{i,t}^\top$ and $\mathbf{V} = \mathbf{V}_{T_0} + \lambda\mathbf{I}$. Let $\widehat{\mathbf{A}}$ be the solution of $\operatorname{argmin}_{\mathbf{A}} \sum_{i=1}^{m} l_i(\mathbf{A})$. Let $[\widehat{\mathbf{A}}]_{k,:}$ and $[\mathbf{A}]_{k,:}$ be the $k$-th rows of $\widehat{\mathbf{A}}$ and $\mathbf{A}$, respectively. Based on Theorem 7, we have with probability at least $(1 - \delta)$,

$$\left\|[\widehat{\mathbf{A}}]_{k,:} - [\mathbf{A}]_{k,:}\right\|_{\mathbf{V}} \le R\sqrt{d\log\left(\frac{1 + mT_0L^2/\lambda}{\delta/n}\right)} + \sqrt{\lambda}L_A, \ \forall k \in [n]. \tag{14}$$

Knowing that $\forall i \in [m]$, $\forall t \in [T_0]$, $\mathbf{x}_{i,t} = (1-\gamma)\mathbf{x}^s + \gamma\zeta_{i,t}$, we have $\lambda_{\max}(\mathbf{x}_{i,t}\mathbf{x}_{i,t}^\top) \le L^2$ and $\mathbb{E}[\mathbf{x}_{i,t}\mathbf{x}_{i,t}^\top] = (1-\gamma)^2\mathbf{x}^s\mathbf{x}^{s\top} + \gamma^2\sigma_\zeta^2\mathbf{I} \succeq \gamma^2\sigma_\zeta^2\mathbf{I}$. Therefore, we have

$$\lambda_{\min}(\mathbb{E}[\mathbf{V}_{T_0}]) = \lambda_{\min}\left(\sum_{i=1}^{m}\sum_{t=1}^{T_0}\mathbb{E}[\mathbf{x}_{i,t}\mathbf{x}_{i,t}^\top]\right) \ge mT_0\gamma^2\sigma_\zeta^2. \tag{15}$$

Based on (15) and Theorem 8, we have

$$\mathbb{P}\left(\lambda_{\min}(\mathbf{V}_{T_0}) \le \epsilon\lambda_{\min}(\mathbb{E}[\mathbf{V}_{T_0}])\right) \le d\exp\left(-(1-\epsilon)^2\frac{mT_0\gamma^2\sigma_\zeta^2}{2L^2}\right). \tag{16}$$

By setting $\epsilon = \frac{1}{2}$ and $T_0 \ge \frac{8L^2}{m\gamma^2\sigma_\zeta^2}\log(\frac{d}{\delta})$, from (16), we have

$$\mathbb{P}\left(\lambda_{\min}(\mathbf{V}) \ge \frac{1}{2}mT_0\gamma^2\sigma_\zeta^2\right) \ge \mathbb{P}\left(\lambda_{\min}(\mathbf{V}_{T_0}) \ge \frac{1}{2}mT_0\gamma^2\sigma_\zeta^2\right) \ge (1-\delta). \tag{17}$$

Combining equations (14) and (17), we have with probability at least $(1 - 2\delta)$,

$$\left\|[\widehat{\mathbf{A}}]_{k,:} - [\mathbf{A}]_{k,:}\right\| \le \frac{R\sqrt{d\log\left(\frac{1+mT_0L^2/\lambda}{\delta/n}\right)} + \sqrt{\lambda}L_A}{\sqrt{\frac{1}{2}m\gamma^2\sigma_\zeta^2T_0}}, \ \forall k \in [n]. \tag{18}$$

Let agent $i$'s local estimate of $\mathbf{A}$ at time $t \in [T_0 + 1, T_0 + T_1]$ returned by the **EXTRA** algorithm (Shi et al., 2015) be denoted by $\widehat{\mathbf{A}}_i^t$. Next, we upper bound the distance between $\widehat{\mathbf{A}} = \operatorname{argmin}_{\mathbf{A}} \sum_{i=1}^{m} l_i(\mathbf{A})$ and $\widehat{\mathbf{A}}_i^t$ based on Theorem 10 as follows. Based on the definition of $l_i(\mathbf{A})$, considering the vectorized version of $\mathbf{A}$, the Hessian matrix has the following expression

$$\nabla^2 l_i(\mathbf{A}) = \sum_{t=1}^{T_0} 2\begin{bmatrix} \mathbf{x}_{i,t}\mathbf{x}_{i,t}^\top & & & \\ & \mathbf{x}_{i,t}\mathbf{x}_{i,t}^\top & & \\ & & \ddots & \\ & & & \mathbf{x}_{i,t}\mathbf{x}_{i,t}^\top \end{bmatrix} + 2\frac{\lambda}{m}\mathbf{I} \preceq 2(T_0L^2 + \frac{\lambda}{m})\mathbf{I},$$

where the inequality is due to the boundedness of the baseline action and the noise vector. From above, we know $\sum_{i=1}^{m} l_i(\mathbf{A}_i)$ is Lipschitz smooth with the constant $2(T_0L^2 + \frac{\lambda}{m})$ and strongly convex with the constant $2\frac{\lambda}{m}$, so by selecting a step size $\alpha < \frac{(\lambda/m)\lambda_{min}(\tilde{\mathbf{P}})}{(T_0L^2+\frac{\lambda}{m})^2}$ as suggested by Theorem 10, there exists a $\tau \in (0,1)$ such that

$$\left\|[\widehat{\mathbf{A}}_i^t]_{k,:} - [\widehat{\mathbf{A}}]_{k,:}\right\| \le \nu\tau^{(t-T_0)}, \ \forall i \in [m], k \in [n], t \in [T_0 + 1, \ldots, T_0 + T_1] \tag{19}$$

where $\nu > 0$ is a constant. Based on (18), (19) and our choice of $T_1$ ($T_1 = (-\log \tau)^{-1} \log(\nu T^\rho)$), for $k \in [n]$, $t \in [T_0 + 1, \ldots, T_0 + T_1]$ and $i, j \in [m]$, we have

$$\left\|[\widehat{\mathbf{A}}_i^t]_{k,:} - [\mathbf{A}]_{k,:}\right\| \leq \left\|[\widehat{\mathbf{A}}_i^t]_{k,:} - [\widehat{\mathbf{A}}]_{k,:}\right\| + \left\|[\widehat{\mathbf{A}}]_{k,:} - [\mathbf{A}]_{k,:}\right\| \leq \frac{1}{T^\rho} + \frac{R\sqrt{d \log\left(\frac{1+mT_0L^2/\lambda}{\delta/n}\right)} + \sqrt{\lambda}L_A}{\sqrt{\frac{1}{2}m\gamma^2\sigma_\zeta^2 T_0}}, \quad (20)$$

and

$$\left\|[\widehat{\mathbf{A}}_i^t]_{k,:} - [\widehat{\mathbf{A}}_j^t]_{k,:}\right\| \leq \left\|[\widehat{\mathbf{A}}_i^t]_{k,:} - [\widehat{\mathbf{A}}]_{k,:}\right\| + \left\|[\widehat{\mathbf{A}}]_{k,:} - [\widehat{\mathbf{A}}_j^t]_{k,:}\right\| \leq \frac{2}{T^\rho}. \quad (21)$$

$\square$

**Lemma 11.** *Define*

$$\mathcal{B}_r \triangleq \frac{1}{T^\rho} + \frac{R\sqrt{d \log\left(\frac{1+mT_0L^2/\lambda}{\delta/n}\right)} + \sqrt{\lambda}L_A}{\sqrt{\frac{1}{2}m\gamma^2\sigma_\zeta^2 T_0}}.$$

*For each agent $i$, construct $\widehat{\mathcal{X}}_i^s$ based on (8) with $\mathcal{C}_{i,k}$ following from (7). By running Algorithm 1 with user-specified $T_0 = \Omega(\frac{L^2}{m\gamma^2\sigma_\zeta^2}\log(\frac{d}{\delta}))$ and $T_1 = \Theta(\log T^\rho)$, there exists a mutual shrunk polytope (see the definition in (12)) subset $\mathcal{X}_{\text{in}}^s$ ($\tau_{\text{in}} = 2\mathcal{B}_r L$) for $\widehat{\mathcal{X}}_i^s$, $\forall i \in [m]$ with probability at least $(1 - 2\delta)$.*

*Proof of Lemma 11.* Consider a mutual shrunk polytope subset $\mathcal{X}_{\text{in}}^s$ ($\tau_{\text{in}} = 2\mathcal{B}_r L$). Based on Lemma 2, with probability at least $1 - 2\delta$, we have for any $\mathbf{x} \in \mathcal{X}_{\text{in}}^s$,

$$
\begin{aligned}
[\widehat{\mathbf{A}}_i]_{k,:}\mathbf{x} + \mathcal{B}_r\|\mathbf{x}\| &= [\mathbf{A}]_{k,:}\mathbf{x} + ([\widehat{\mathbf{A}}_i]_{k,:} - [\mathbf{A}]_{k,:})\mathbf{x} + \mathcal{B}_r\|\mathbf{x}\| \\
&\leq [\mathbf{A}]_{k,:}\mathbf{x} + \left\|[\widehat{\mathbf{A}}_i]_{k,:} - [\mathbf{A}]_{k,:}\right\|\|\mathbf{x}\| + \mathcal{B}_r\|\mathbf{x}\| \\
&\leq [\mathbf{A}]_{k,:}\mathbf{x} + 2\mathcal{B}_r\|\mathbf{x}\| \leq [\mathbf{A}]_{k,:}\mathbf{x} + 2\mathcal{B}_r L \leq b_k, \ \forall k \in [n] \text{ and } \forall i \in [m],
\end{aligned}
\quad (22)
$$

which implies that $\mathcal{X}_{\text{in}}^s \subset \widehat{\mathcal{X}}_i^s$, $\forall i$. $\square$

**Lemma 12.** *For each agent $i$, construct $\widehat{\mathcal{X}}_i^s$ based on (8) with $\mathcal{C}_{i,k}$ following from (7). By running Algorithm 1 with user-specified $T_0 = \Omega(\frac{L^2}{m\gamma^2\sigma_\zeta^2}\log(\frac{d}{\delta}))$ and $T_1 = \Theta(\log T^\rho)$, we have for any point $\mathbf{x}$,*

$$\left\|\Pi_{\widehat{\mathcal{X}}_i^s}(\mathbf{x}) - \Pi_{\widehat{\mathcal{X}}_j^s}(\mathbf{x})\right\| \leq O(\frac{1}{T^\rho}), \ \forall i, j \in [m]. \quad (23)$$

Before we discuss the proof of Lemma 12, for the sake of completeness, we provide the formal statement of Theorem 3.1 in Bonnans et al. (1998), used in the derivation of Lemma 12.

We first define the notations used in (Bonnans et al., 1998). Note that the notations here are only locally defined for the statement of Theorem 3.1 in Bonnans et al. (1998). The work of Bonnans et al. (1998) focuses on the sensitivity analysis of parametric optimization problems of the form

$$(P_{\mathbf{u}}) : \min_{\mathbf{x} \in \mathcal{X}} f(\mathbf{x}, \mathbf{u}) \text{ subject to } G(\mathbf{x}, \mathbf{u}) \in \mathcal{K},$$

where $\mathcal{X}$ is a finite dimensional space, $\mathcal{U}$ is a Banach space, $\mathcal{K}$ is a closed subset of Banach space $\mathcal{Y}$ and $f$ and $G$ are twice continuously differentiable mappings from $\mathcal{X} \times \mathcal{U}$ to R and $\mathcal{Y}$, respectively. The optimization problem is considered to be unperturbed when $\mathbf{u} = 0$.

Given $\mathbf{u}$, the feasible set, optimal value and set of optimal solutions of $(P_{\mathbf{u}})$ are denoted as follows

$$
\begin{aligned}
\Phi(\mathbf{u}) &\triangleq \{\mathbf{x} \in \mathcal{X} : G(\mathbf{x}, \mathbf{u}) \in \mathcal{K}\}, \\
v(\mathbf{u}) &\triangleq \inf\{f(\mathbf{x}, \mathbf{u}) : \mathbf{x} \in \Phi(\mathbf{u})\}, \\
S(\mathbf{u}) &\triangleq \operatorname{argmin}\{f(\mathbf{x}, \mathbf{u}) : \mathbf{x} \in \Phi(\mathbf{u})\}.
\end{aligned}
$$

A point $\mathbf{x} \in \mathcal{X}$ is called an $\epsilon$-optimal solution of $(P_\mathbf{u})$ if $\mathbf{x} \in \Phi(\mathbf{u})$ and $f(\mathbf{x}, \mathbf{u}) \leq v(\mathbf{u}) + \epsilon$.

We also define the following notations to present the theorem statement.

| | |
|---|---|
| $\mathcal{Y}^*$ | Dual space of $\mathcal{Y}$ |
| $\text{dist}(\mathbf{y}, \mathcal{X})$ | The minimum distance from point $\mathbf{y}$ to set $\mathcal{X}$: $\inf\{\|\mathbf{y} - \mathbf{x}\| : \mathbf{x} \in \mathcal{X}\}$ |
| $T_\mathcal{K}(\mathbf{y})$ | The tangent cone to the set $\mathcal{K}$ at the point $\mathbf{y} \in \mathcal{K}$: $\{\mathbf{h} \in \mathcal{Y} : \text{dist}(\mathbf{y} + t\mathbf{h}, \mathcal{K}) = o(t)\}$ |
| $N_\mathcal{K}(\mathbf{y})$ | The normal cone to the set $\mathcal{K}$ at the point $\mathbf{y} \in \mathcal{K}$: $\{\mathbf{y}^* \in \mathcal{Y}^* : \langle \mathbf{y}^*, \mathbf{h} \rangle \leq 0, \forall \mathbf{h} \in T_\mathcal{K}(\mathbf{y})\}$ |
| $Df(\mathbf{x}, \mathbf{u})$ | Derivative of $f$ |
| $D_\mathbf{x} f(\mathbf{x}, \mathbf{u})$ | Partial derivative of $f$ w.r.t. $\mathbf{x}$ |
| $D_{\mathbf{xx}} f(\mathbf{x}, \mathbf{u})$ | Second order derivative of $f$ w.r.t. $\mathbf{x}$ |
| $Df(\mathbf{x}', \mathbf{u}')(\mathbf{x}, \mathbf{u})$ | The linear function based on the derivative at $(\mathbf{x}', \mathbf{u}')$ |
| $L(\mathbf{x}, \lambda, \mathbf{u})$ | The Lagrangian $f(\mathbf{x}, \mathbf{u}) + \langle \lambda, G(\mathbf{x}, \mathbf{u}) \rangle$, $\lambda \in \mathcal{Y}^*$ |
| $\Lambda_\mathbf{u}(\mathbf{x})$ | $\{\lambda \in N_\mathcal{K}(G(\mathbf{x}, \mathbf{u})) : D_\mathbf{x} L(\mathbf{x}, \lambda, \mathbf{u}) = 0\}$ |
| $\{\mathcal{X}_1 + \mathcal{X}_2\}$ | $\cup\{\mathbf{x}_1 + \mathbf{x}_2\}$, $\mathbf{x}_1 \in \mathcal{X}_1, \mathbf{x}_2 \in \mathcal{X}_2$ |
| $\text{int}(\mathcal{X})$ | The interior of the set $\mathcal{X}$ |

To study the first order differentiabilitiy of the optimal value function $v(\mathbf{u})$, for a given direction $\mathbf{d} \in \mathcal{U}$ and the optimal solution of the unperturbed problem $\mathbf{x}_0 \in \mathcal{S}(0)$, Bonnans et al. (1998) consider the linearization of the family of problems $(P_{t\mathbf{d}})$ and its dual as follows

$$(PL_\mathbf{d}) : \min_\mathbf{h} f(\mathbf{x}_0, 0)(\mathbf{h}, \mathbf{d}) \text{ subject to } DG(\mathbf{x}_0, 0)(\mathbf{h}, \mathbf{d}) \in T_\mathcal{K}(G(\mathbf{x}_0, 0)),$$

$$(DL_\mathbf{d}) : \max_{\lambda \in \Lambda_0(\mathbf{x}_0)} D_\mathbf{u} L(\mathbf{x}_0, \lambda, 0)\mathbf{d}.$$

**Theorem 13.** *(Theorem 3.1 in Bonnans et al. (1998)) Let $\bar{\mathbf{x}}(t)$ be an $O(t^2)$-optimal trajectory of $(P_{t\mathbf{d}})$ converging to a point $\mathbf{x}_0 \in \Phi(0)$ as $t \to 0$. Assume $v(PL_\mathbf{d})$ to be finite. Suppose that the following conditions hold:*

1. *$\mathbf{x}_0$ satisfies the directional constraint qualification, which is implied if*

$$0 \in int\{G(\mathbf{x}_0, 0) + D_\mathbf{x} G(\mathbf{x}_0, 0)\mathcal{X} - \mathcal{K}\}.$$

2. *$v(t\mathbf{d}) \leq v(0) + tv(PL_\mathbf{d}) + O(t^2)$, $t \geq 0$ (Equation 3.4 in (Bonnans et al., 1998)).*

3. *The strong second order sufficient condition (Equation 3.1 in (Bonnans et al., 1998)) holds, which is implied if*

$$\sup_{\lambda \in \mathcal{S}(DL_\mathbf{d})} D^2_{\mathbf{xx}} L(\mathbf{x}_0, \lambda, 0)(\mathbf{h}, \mathbf{h}) > 0, \ \forall \mathbf{h} \in C(\mathbf{x}_0) \backslash \{0\},$$

   *where $C(\mathbf{x}_0)$ denotes the critical cone.*

*Then $\bar{\mathbf{x}}(t)$ is Lipschitz stable at $\mathbf{x}_0$, i.e., for $t \geq 0$, $\|\bar{\mathbf{x}}(t) - \mathbf{x}_0\| = O(t)$.*

*Proof.* (Proof of Lemma 12) The key idea is to leverage Theorem 13, which quantifies the sensitivity of the optimal solution of a "perturbed" optimization problem. More specifically, it is shown that the distance between the original optimal solution and the optimal solution of the perturbed problem is upper-bounded by the magnitude of the perturbation.

First, we show that $\forall i \in [m]$, the projection problem $\Pi_{\widehat{\mathcal{X}}_i^s}(\mathbf{x})$ can be formulated as a quadratic programming with second-order cone constraints. The definition of $\widehat{\mathcal{X}}_i^s$ has the following equivalent expression

$$\widehat{\mathcal{X}}_i^s \triangleq \{\mathbf{x} \in \mathrm{R}^d : \tilde{\mathbf{a}}_k^\top \mathbf{x} \leq b_k, \ \forall \tilde{\mathbf{a}}_k \in \mathcal{C}_{i,k}, \ \forall k \in [n]\} = \{\mathbf{x} \in \mathrm{R}^d : \max_{\tilde{\mathbf{a}}_k \in \mathcal{C}_{i,k}} \tilde{\mathbf{a}}_k^\top \mathbf{x} \leq b_k, \ \forall k \in [n]\}$$

$$= \{\mathbf{x} \in \mathrm{R}^d : [\widehat{\mathbf{A}}_i]_{k,:}\mathbf{x} + \mathcal{B}_r\|\mathbf{x}\| \leq b_k, \ \forall k \in [n]\}, \tag{24}$$

where each second-order cone inequality: $[\widehat{\mathbf{A}}_i]_{k,:}\mathbf{x} + \mathcal{B}_r\|\mathbf{x}\| \leq b_k$ can be equivalently written as a linear matrix inequality (LMI):

$$[\widehat{\mathbf{A}}_i]_{k,:}\mathbf{x} + \mathcal{B}_r\|\mathbf{x}\| \leq b_k \Leftrightarrow \mathbf{G}^k(\mathbf{x}, \widehat{\mathbf{A}}_i) \triangleq \begin{bmatrix} (b_k - [\widehat{\mathbf{A}}_i]_{k,:}\mathbf{x}) & \mathcal{B}_r\mathbf{x}^\top \\ \mathcal{B}_r\mathbf{x} & (b_k - [\widehat{\mathbf{A}}_i]_{k,:}\mathbf{x})\mathbf{I} \end{bmatrix} \succeq 0. \tag{25}$$

For simplicity, we define the following matrix

$$\mathcal{G}(\mathbf{x}, \widehat{\mathbf{A}}_i) \triangleq \begin{bmatrix} \mathbf{G}^1(\mathbf{x}, \widehat{\mathbf{A}}_i) & & & \\ & \mathbf{G}^2(\mathbf{x}, \widehat{\mathbf{A}}_i) & & \\ & & \ddots & \\ & & & \mathbf{G}^n(\mathbf{x}, \widehat{\mathbf{A}}_i) \end{bmatrix}.$$

Considering the intersection of all LMIs, we have

$$\widehat{\mathcal{X}}_i^s \triangleq \{\mathbf{x} \in \mathrm{R}^d : \mathcal{G}(\mathbf{x}, \widehat{\mathbf{A}}_i) \succeq 0\}. \tag{26}$$

Based on (26), for a point $\mathbf{x} \in \mathrm{R}^d$, we have $\Pi_{\widehat{\mathcal{X}}_i^s}(\mathbf{x}) = \mathbf{x} + \xi_i$, where $\xi_i$ is derived by solving the following optimization problem

$$\xi_i = \mathrm{argmin}_\xi \; \xi^\top \xi, \; \text{s.t.} \begin{bmatrix} \mathbf{G}^1(\mathbf{x}+\xi, \widehat{\mathbf{A}}_i) & & & \\ & \mathbf{G}^2(\mathbf{x}+\xi, \widehat{\mathbf{A}}_i) & & \\ & & \ddots & \\ & & & \mathbf{G}^n(\mathbf{x}+\xi, \widehat{\mathbf{A}}_i) \end{bmatrix} \succeq 0. \tag{27}$$

Based on Lemma 2, we have $\|[\widehat{\mathbf{A}}_i]_{k,:} - [\widehat{\mathbf{A}}_j]_{k,:}\| = O(\frac{1}{T^\rho})$, $\forall i,j \in [m]$ and $\forall k \in [n]$. Therefore, $[\widehat{\mathbf{A}}_j]_{k,:}$ can be expressed as $[\widehat{\mathbf{A}}_i]_{k,:} + \psi_k$, where $\|\psi_k\| = O(\frac{1}{T^\rho})$. With this expression, the projection $\Pi_{\widehat{\mathcal{X}}_j^s}(\mathbf{x}) = \mathbf{x} + \xi_j$ can be formulated as a perturbed version of the optimization (27), where the perturbation is parameterized in terms of $\psi = [\psi_1, \ldots, \psi_n]$ as follows:

$$\xi_j = \mathrm{argmin}_\xi \; \xi^\top \xi, \; \text{s.t.} \begin{bmatrix} \mathbf{G}^1(\mathbf{x}+\xi, \widehat{\mathbf{A}}_i+\psi) & & & \\ & \mathbf{G}^2(\mathbf{x}+\xi, \widehat{\mathbf{A}}_i+\psi) & & \\ & & \ddots & \\ & & & \mathbf{G}^n(\mathbf{x}+\xi, \widehat{\mathbf{A}}_i+\psi) \end{bmatrix} \succeq 0. \tag{28}$$

To show that $\|\Pi_{\widehat{\mathcal{X}}_i^s}(\mathbf{x}) - \Pi_{\widehat{\mathcal{X}}_j^s}(\mathbf{x})\| = \|\xi_i - \xi_j\| = O(\|\psi\|) = O(\frac{1}{T^\rho})$, we apply Theorem 13, where three conditions need to be satisfied: directional constraint qualification (DCQ), Equation 3.4 in Bonnans et al. (1998) and strong second-order sufficient conditions (we refer readers to Bonnans et al. (1998) for detailed definitions).

- **DCQ**:
  A sufficient condition for DCQ is constraint qualification (CQ) (see the definition in Bonnans et al. (1998)), which is satisfied in our problem formulation if the first-order approximation of $\mathcal{G}(\mathbf{x}+\xi, \widehat{\mathbf{A}}_i+\psi)$ w.r.t. the variable $\xi$ can be positive-definite. Noting that $\mathcal{G}(\mathbf{x}+\xi, \widehat{\mathbf{A}}_i+\psi)$ is an affine function of $\xi$, the first-order approximation is exactly the original function. Now suppose that $\forall i \in [m]$, $\widehat{\mathcal{X}}_i^s$ has a strictly feasible point (this is implied by the existence of the mutual shrunk polytope), which means there exists a $\hat{\xi}$ such that $\mathcal{G}(\mathbf{x}+\hat{\xi}, \widehat{\mathbf{A}}_i+\psi)$ is positive-definite, and then CQ is satisfied.

- **Equation 3.4 in Bonnans et al. (1998)**:
  In Bonnans et al. (1998), the authors provided the sufficient conditions for Equation 3.4: DCQ and second-order regularity (Definition 2.2 in Bonnans et al. (1998)). DCQ, as mentioned previously, holds in our case, and second-order regularity holds for semi-definite optimization, which is the case for our problem setup.

- **Second-order sufficient conditions**:
  The strong second-order sufficient condition (Equation 3.1 in Bonnans et al. (1998)) has an alternative form (Equation 3.3 in Bonnans et al. (1998)), which is satisfied in our problem setup since the Hessian of the Lagrangian is $2\mathbf{I}$, which is positive-definite.

Since all the conditions above are met, the lemma is proved by applying Theorem 13. □

## A.3 Convex Part

**Lemma 14.** *Let Algorithm 2 run with step size $\eta > 0$ and define $\mathbf{x}_t \triangleq \frac{1}{m} \sum_{i=1}^{m} \mathbf{x}_{i,t}$ and $\mathbf{y}_t \triangleq \frac{1}{m} \sum_{i=1}^{m} \mathbf{y}_{i,t}$. Under Assumptions 1 to 3 and the fact that gradients are bounded, i.e., $\left\|\nabla f_{i,t}(\mathbf{x})\right\| \leq G$ for any $\mathbf{x} \in \mathcal{X}^s$, we have that $\forall i \in [m]$*

$$\left\|\mathbf{x}_t - \mathbf{x}_{i,t}\right\| \leq \left(O(\frac{1}{T^\rho}) + 2\eta G\right)\frac{\sqrt{m}\beta}{1-\beta}.$$

*Proof.* For the presentation simplicity, we define the following matrices

$$\mathbf{X}_t \triangleq [\mathbf{x}_{1,t}, \ldots, \mathbf{x}_{m,t}], \ \mathbf{Y}_t \triangleq [\mathbf{y}_{1,t}, \ldots, \mathbf{y}_{m,t}], \ \mathbf{G}_t \triangleq [\nabla f_{1,t}(\mathbf{x}_{1,t}), \ldots, \nabla f_{m,t}(\mathbf{x}_{m,t})], \text{ and } \mathbf{R}_t \triangleq [\mathbf{r}_{1,t}, \ldots, \mathbf{r}_{m,t}],$$

where $\mathbf{r}_{i,t} \triangleq \mathbf{y}_{i,t} - \left(\mathbf{x}_{i,t} - \eta\nabla f_{i,t}(\mathbf{x}_{i,t})\right)$. Then, the update can be expressed as $\mathbf{X}_t = \mathbf{Y}_{t-1}\mathbf{P} = \left(\mathbf{X}_{t-1} - \eta\mathbf{G}_{t-1} + \mathbf{R}_{t-1}\right)\mathbf{P}$.

Expanding the update recursively, we have

$$\mathbf{X}_t = \mathbf{X}_{T_s}\mathbf{P}^{(t-T_s)} - \eta\sum_{l=1}^{t-T_s}\mathbf{G}_{t-l}\mathbf{P}^l + \sum_{l=1}^{t-T_s}\mathbf{R}_{t-l}\mathbf{P}^l. \tag{29}$$

Since $\mathbf{P}$ is doubly stochastic, we have $\mathbf{P}^k\mathbf{1} = \mathbf{1}$ for all $k \geq 1$. Based on the geometric mixing bound of $\mathbf{P}$ and the above equation we get

$$\left\|\mathbf{x}_t - \mathbf{x}_{i,t}\right\| = \left\|\mathbf{X}_t(\frac{1}{m}\mathbf{1} - \mathbf{e}_i)\right\|$$

$$\leq \left\|\mathbf{x}_{T_s} - \mathbf{X}_{T_s}[\mathbf{P}^{(t-T_s)}]_{:,i}\right\| + \eta\sum_{l=1}^{t-T_s}\left\|\mathbf{G}_{t-l}(\frac{1}{m}\mathbf{1} - [\mathbf{P}^l]_{:,i})\right\| + \sum_{l=1}^{t-T_s}\left\|\mathbf{R}_{t-l}(\frac{1}{m}\mathbf{1} - [\mathbf{P}^l]_{:,i})\right\|$$

$$\leq \sum_{l=1}^{t-T_s}(\eta G)\sqrt{m}\beta^l + \sum_{l=1}^{t-T_s}\left(O(\frac{1}{T^\rho}) + \eta G\right)\sqrt{m}\beta^l$$

$$\leq \left(O(\frac{1}{T^\rho}) + 2\eta G\right)\frac{\sqrt{m}\beta}{1-\beta},$$

where $\left\|\mathbf{x}_{T_s} - \mathbf{X}_{T_s}[\mathbf{P}^{t-T_s}]_{:,i}\right\| = 0$ by the identical initialization of all agents with the same action at $T_s$, and the other inequality is based on Lemma 12 as follows

$$\left\|\mathbf{r}_{i,t}\right\| = \left\|\mathbf{y}_{i,t} - \left(\mathbf{x}_{i,t} - \eta\nabla f_{i,t}(\mathbf{x}_{i,t})\right)\right\|$$

$$\leq \left\|\sum_j[\mathbf{P}]_{ji}\Pi_{\widehat{\mathcal{X}}_i^s}[\mathbf{y}_{j,t-1}] - \left(\sum_j[\mathbf{P}]_{ji}\mathbf{y}_{j,t-1} - \eta\nabla f_{i,t}(\mathbf{x}_{i,t})\right)\right\|$$

$$\leq O(T^{-\rho}) + \eta G.$$

□

*Proof of Theorem 3.* First, we decompose the individual regret of agent $j$ into three terms:

$$\sum_t \sum_i f_{i,t}(\mathbf{x}_{j,t}) - \sum_t f_t(\mathbf{x}_t^*) = \underbrace{\sum_{t=1}^{T_s-1} \sum_i f_{i,t}(\mathbf{x}_{j,t}) - f_{i,t}(\mathbf{x}_t^*)}_{\text{Term I}} + \underbrace{\sum_{t=T_s}^{T} \sum_i f_{i,t}(\mathbf{x}_{j,t}) - f_{i,t}(\tilde{\mathbf{x}}_t^*)}_{\text{Term II}} + \underbrace{\sum_{t=T_s}^{T} f_t(\tilde{\mathbf{x}}_t^*) - f_t(\mathbf{x}_t^*)}_{\text{Term III}},$$

$$(30)$$

where $\tilde{\mathbf{x}}_t^*$ is the projection of $\mathbf{x}_t^*$ on $\mathcal{X}_{\text{in}}^s$, which is a mutual subset of $\{\widehat{\mathcal{X}}_i^s\}_{i\in[m]}$ with $\tau_{\text{in}} = 2\mathcal{B}_r L$ based on Equation (22) in Lemma 11. We now proceed to bound each term.

**The upper bound of Term I**:
Note that by choosing $\gamma \le \frac{\Delta^s}{LL_A}$, we have $\forall i \in [m]$ and $t \in [1, \ldots, T_0 + T_1]$

$$[\mathbf{A}]_{k,:}\mathbf{x}_{i,t} = [\mathbf{A}]_{k,:}\left((1-\gamma)\mathbf{x}^s + \gamma\zeta_{i,t}\right) \le (1-\gamma)b_k^s + \Delta^s \le (1-\gamma)b_k^s + (b_k - b_k^s) < b_k, \tag{31}$$

which implies the safeness of the action.

Based on the Lipschitz property of the function sequence, we have

$$\sum_{t=1}^{T_s-1} \sum_i f_{i,t}(\mathbf{x}_{j,t}) - f_{i,t}(\mathbf{x}_t^*) \le \sum_{t=1}^{T_s-1} \sum_i G\|\mathbf{x}_{j,t} - \mathbf{x}_t^*\| \le 2GLm(T_0 + T_1). \tag{32}$$

**The upper bound of Term II**:
Based on the update rule, $\forall i \in [m]$ and $t \in [T_s, \ldots, T]$ we have

$$\begin{aligned}
f_{i,t}(\mathbf{x}_{i,t}) - f_{i,t}(\tilde{\mathbf{x}}_t^*) \le& \nabla f_{i,t}(\mathbf{x}_{i,t})^\top (\mathbf{x}_{i,t} - \tilde{\mathbf{x}}_t^*) \\
=& \frac{1}{\eta}\left[\frac{1}{2}\eta^2\|\nabla f_{i,t}(\mathbf{x}_{i,t})\|^2 + \frac{1}{2}\|\mathbf{x}_{i,t} - \tilde{\mathbf{x}}_t^*\|^2 - \frac{1}{2}\|\mathbf{x}_{i,t} - \tilde{\mathbf{x}}_t^* - \eta\nabla f_{i,t}(\mathbf{x}_{i,t})\|^2\right] \\
\le& \frac{1}{\eta}\left[\frac{1}{2}\eta^2\|\nabla f_{i,t}(\mathbf{x}_{i,t})\|^2 + \frac{1}{2}\|\mathbf{x}_{i,t} - \tilde{\mathbf{x}}_t^*\|^2 - \frac{1}{2}\|\mathbf{y}_{i,t} - \tilde{\mathbf{x}}_t^*\|^2\right] \\
=& \frac{1}{\eta}\left[\frac{1}{2}\eta^2\|\nabla f_{i,t}(\mathbf{x}_{i,t})\|^2 + \frac{1}{2}\Big\|\sum_j[\mathbf{P}]_{ji}\mathbf{y}_{j,t-1} - \tilde{\mathbf{x}}_t^*\Big\|^2 - \frac{1}{2}\|\mathbf{y}_{i,t} - \tilde{\mathbf{x}}_t^*\|^2\right] \\
\le& \frac{1}{\eta}\left[\frac{1}{2}\eta^2\|\nabla f_{i,t}(\mathbf{x}_{i,t})\|^2 + \frac{1}{2}\sum_j[\mathbf{P}]_{ji}\|\mathbf{y}_{j,t-1} - \tilde{\mathbf{x}}_t^*\|^2 - \frac{1}{2}\|\mathbf{y}_{i,t} - \tilde{\mathbf{x}}_t^*\|^2\right],
\end{aligned} \tag{33}$$

where the second inequality is due to the projection property that $\|\mathbf{y}_{i,t} - \tilde{\mathbf{x}}_t^*\| \le \|\mathbf{x}_{i,t} - \eta\nabla f_{i,t}(\mathbf{x}_{i,t}) - \tilde{\mathbf{x}}_t^*\|$, and the third inequality is due to the convexity of the square function.

Based on Equation (33) and Lemma 14, we have

$$\begin{aligned}
f_{i,t}(\mathbf{x}_{j,t}) - f_{i,t}(\tilde{\mathbf{x}}_t^*) =& f_{i,t}(\mathbf{x}_{j,t}) - f_{i,t}(\mathbf{x}_{i,t}) + f_{i,t}(\mathbf{x}_{i,t}) - f_{i,t}(\tilde{\mathbf{x}}_t^*) \\
\le& G\|\mathbf{x}_{j,t} - \mathbf{x}_{i,t}\| + f_{i,t}(\mathbf{x}_{i,t}) - f_{i,t}(\tilde{\mathbf{x}}_t^*) \\
\le& 2G\big(O(\frac{1}{T^\rho}) + 2\eta G\big)\frac{\sqrt{m}\beta}{1-\beta} + \frac{1}{2}\eta\|\nabla f_{i,t}(\mathbf{x}_{i,t})\|^2 + \frac{1}{2\eta}\sum_j[\mathbf{P}]_{ji}\|\mathbf{y}_{j,t-1} - \tilde{\mathbf{x}}_t^*\|^2 - \frac{1}{2\eta}\|\mathbf{y}_{i,t} - \tilde{\mathbf{x}}_t^*\|^2.
\end{aligned} \tag{34}$$

Summing Equation (34) over $i$, we get

$$\sum_i \left(f_{i,t}(\mathbf{x}_{j,t}) - f_{i,t}(\tilde{\mathbf{x}}_t^*)\right)$$

$$\leq 2mG\left(O(\frac{1}{T^\rho}) + 2\eta G\right)\frac{\sqrt{m}\beta}{1-\beta} + \frac{\eta}{2}\sum_i \left\|\nabla f_{i,t}(\mathbf{x}_{i,t})\right\|^2 + \frac{1}{2\eta}\sum_j \left\|\mathbf{y}_{j,t-1} - \tilde{\mathbf{x}}_t^*\right\|^2 - \frac{1}{2\eta}\sum_i \left\|\mathbf{y}_{i,t} - \tilde{\mathbf{x}}_t^*\right\|^2$$

$$= 2mG\left(O(\frac{1}{T^\rho}) + 2\eta G\right)\frac{\sqrt{m}\beta}{1-\beta} + \frac{\eta}{2}\sum_i \left\|\nabla f_{i,t}(\mathbf{x}_{i,t})\right\|^2 + \frac{1}{2\eta}\sum_i \left(\left\|\mathbf{y}_{i,t-1}\right\|^2 - \left\|\mathbf{y}_{i,t}\right\|^2 + 2(\mathbf{y}_{i,t} - \mathbf{y}_{i,t-1})^\top \tilde{\mathbf{x}}_t^*\right).$$

$$(35)$$

Summing Equation (35) over $t \in [T_s, \dots, T]$, we have

$$\sum_{t=T_s}^{T}\sum_i \left(f_{i,t}(\mathbf{x}_{j,t}) - f_{i,t}(\tilde{\mathbf{x}}_t^*)\right)$$

$$\leq \frac{\eta}{2}\sum_{t=T_s}^{T}\sum_i \left\|\nabla f_{i,t}(\mathbf{x}_{i,t})\right\|^2 + \frac{1}{2\eta}\sum_i \left\|\mathbf{y}_{i,T_s-1}\right\|^2 + \frac{1}{\eta}\left(\sum_i \mathbf{y}_{i,T}^\top \tilde{\mathbf{x}}_T^* - \sum_i \mathbf{y}_{i,T_s-1}^\top \tilde{\mathbf{x}}_{T_s-1}^*\right)$$

$$+ \frac{1}{\eta}\sum_{t=T_s-1}^{T-1}\sum_i (\tilde{\mathbf{x}}_t^* - \tilde{\mathbf{x}}_{t+1}^*)^\top \mathbf{y}_{i,t} + 2TmG\left(O(\frac{1}{T^\rho}) + 2\eta G\right)\frac{\sqrt{m}\beta}{1-\beta}.$$

$$(36)$$

**The upper bound of Term III**:

Based on Lemma 9, we have for any $\mathbf{x}_t^* \in \mathcal{X}^s$ and its projection to $\mathcal{X}_{\text{in}}^s$, denoted by $\tilde{\mathbf{x}}_t^*$, that

$$\sum_{t=T_s}^{T}\sum_i \left(f_{i,t}(\tilde{\mathbf{x}}_t^*) - f_{i,t}(\mathbf{x}_t^*)\right) \leq \sum_{t=T_s}^{T}\sum_i G\left\|\tilde{\mathbf{x}}_t^* - \mathbf{x}_t^*\right\| \leq mTG\frac{2\sqrt{d}L\mathcal{B}_r}{C(\mathbf{A}, \mathbf{b})}.$$

$$(37)$$

Substituting Equations (32), (36) and (37) into Equation (30), we get

$$\sum_t\sum_i \left(f_{i,t}(\mathbf{x}_{j,t}) - f_{i,t}(\mathbf{x}_t^*)\right)$$

$$\leq 2GLm(T_0 + T_1) + \frac{\eta mTG^2}{2} + \frac{1}{2\eta}\sum_i \left\|\mathbf{y}_{i,T_s-1}\right\|^2 + \frac{1}{\eta}\left(\sum_i \mathbf{y}_{i,T}^\top \tilde{\mathbf{x}}_T^* - \sum_i \mathbf{y}_{i,T_s-1}^\top \tilde{\mathbf{x}}_{T_s-1}^*\right)$$

$$+ \frac{1}{\eta}\sum_{t=T_s-1}^{T-1}\sum_i (\tilde{\mathbf{x}}_t^* - \tilde{\mathbf{x}}_{t+1}^*)^\top \mathbf{y}_{i,t} + 2TmG\left(O(\frac{1}{T^\rho}) + 2\eta G\right)\frac{\sqrt{m}\beta}{1-\beta} + mTG\frac{2\sqrt{d}L\mathcal{B}_r}{C(\mathbf{A}, \mathbf{b})},$$

$$(38)$$

which is $O(T_0 + T_1 + \frac{1}{\eta} + \frac{1}{\eta}C_T^* + \frac{T\sqrt{\log T_0}}{\sqrt{T_0}} + \frac{\beta\eta T}{(1-\beta)})$ and the final regret bound is derived by substituting the choices of $\eta$ and $T_0$ into above. $\qquad\square$

## A.4 Non-convex Part

**Lemma 15** (Lemma 4 in Ghai et al. (2022)). *Suppose Assumptions 5, 6, 7 hold and $\mathbf{u}_t = q(\mathbf{x}_t)$, then $\left\|q(\mathbf{x}_{t+1}) - \mathbf{u}_{t+1}\right\| = O(W^4 G_F^{3/2}\eta^{3/2})$ based on the following update rule:*

$$\mathbf{u}_{t+1} = argmin_{\mathbf{u} \in \mathcal{X}^{s\prime}}\left\{\nabla \tilde{f}_t(\mathbf{u}_t)^\top \mathbf{u} + \frac{1}{\eta}\mathcal{D}_\phi(\mathbf{u}, \mathbf{u}_t)\right\},$$

$$\mathbf{x}_{t+1} = argmin_{\mathbf{x} \in \mathcal{X}^s}\left\{\nabla f_t(\mathbf{x}_t)^\top \mathbf{x} + \frac{1}{2\eta}\left\|\mathbf{x} - \mathbf{x}_t\right\|^2\right\}.$$

**Theorem 16** (Theorem 7 in Ghai et al. (2022)). *Given a convex and compact domain $\mathcal{X} \subset \mathcal{X}^s$, and not necessarily convex loss $f_t(\cdot)$ satisfying Assumption 7. When Assumption 8 is met, there exists an OMD object with convex loss $\tilde{f}_t(\cdot)$, a convex domain and a strongly convex regularization $\phi$ satisfying Assumption 5.*

**Lemma 17.** *Suppose Assumptions 5-7 hold and $\mathbf{u}_{i,t} = q(\mathbf{x}_{i,t})$, $\forall i \in [m]$; then*

$$\left\| q(\mathbf{x}_{i,t+1}) - \mathbf{u}'_{i,t+1} \right\| = O(\frac{1}{T^{2\rho}} + \eta^{3/2}),$$

*based on the following update rules:*

$$
\begin{aligned}
\mathbf{z}_{i,t} &= argmin_{\mathbf{u} \in \widehat{X}_i^{s'}} \left\{ \nabla \tilde{f}_{i,t}(\mathbf{u}_{i,t})^\top \mathbf{u} + \frac{1}{\eta} \mathcal{D}_\phi(\mathbf{u}, \mathbf{u}_{i,t}) \right\}, \\
\mathbf{u}'_{i,t+1} &= \sum_j [\mathbf{P}]_{ji} \mathbf{z}_{j,t}, \\
\mathbf{y}_{i,t} &= argmin_{\mathbf{x} \in \widehat{X}_i^s} \left\{ \nabla f_{i,t}(\mathbf{x}_{i,t})^\top \mathbf{x} + \frac{1}{2\eta} \left\| \mathbf{x} - \mathbf{x}_{i,t} \right\|^2 \right\}, \\
\mathbf{x}_{i,t+1} &= \sum_j [\mathbf{P}]_{ji} \mathbf{y}_{j,t}.
\end{aligned}
\tag{39}
$$

*Proof.* We first upper bound $\left\| q(\mathbf{x}_{i,t+1}) - \mathbf{u}'_{i,t+1} \right\|$ as follows

$$\left\| q(\mathbf{x}_{i,t+1}) - \mathbf{u}'_{i,t+1} \right\| \leq \left\| \sum_j [\mathbf{P}]_{ji} \mathbf{z}_{j,t} - \sum_j [\mathbf{P}]_{ji} q(\mathbf{y}_{j,t}) \right\| + \left\| \sum_j [\mathbf{P}]_{ji} q(\mathbf{y}_{j,t}) - q(\sum_j [\mathbf{P}]_{ji} \mathbf{y}_{j,t}) \right\|. \tag{40}$$

To bound the second term, we consider the Taylor expansion of $q(\mathbf{y})$ w.r.t. a point $\hat{\mathbf{y}}$ in the convex hull of $\{\mathbf{y}_{i,t}\}_i$:

$$
\begin{aligned}
\left\| \sum_j [\mathbf{P}]_{ji} q(\mathbf{y}_{j,t}) - q(\sum_j [\mathbf{P}]_{ji} \mathbf{y}_{j,t}) \right\| \leq & \left\| \sum_j [\mathbf{P}]_{ji} \left( q(\hat{\mathbf{y}}) + J_q(\hat{\mathbf{y}})(\mathbf{y}_{j,t} - \hat{\mathbf{y}}) + O(\left\| \mathbf{y}_{j,t} - \hat{\mathbf{y}} \right\|^2) \right) \right. \\
& \left. - \left( q(\hat{\mathbf{y}}) + J_q(\hat{\mathbf{y}})(\sum_j [\mathbf{P}]_{ji} \mathbf{y}_{j,t} - \hat{\mathbf{y}}) + O(\left\| \sum_j [\mathbf{P}]_{ji} \mathbf{y}_{j,t} - \hat{\mathbf{y}} \right\|^2) \right) \right\| \\
\leq & O\left( \sum_j [\mathbf{P}]_{ji} \left\| \mathbf{y}_{j,t} - \hat{\mathbf{y}} \right\|^2 \right) + O\left( \left\| \sum_j [\mathbf{P}]_{ji} \mathbf{y}_{j,t} - \hat{\mathbf{y}} \right\|^2 \right) \\
\leq & O(D^2),
\end{aligned}
\tag{41}
$$

where $D$ denotes the diameter of the convex hull of $\{\mathbf{y}_{i,t}\}$ and is upper bounded as follows

$$
\begin{aligned}
D \triangleq & \max_{(i,j)} \left\| \mathbf{y}_{i,t} - \mathbf{y}_{j,t} \right\| \\
= & \max_{(i,j)} \left\| \Pi_{\widehat{X}_i^s}\left( \mathbf{x}_{i,t} - \eta \nabla f_{i,t}(\mathbf{x}_{i,t}) \right) - \Pi_{\widehat{X}_j^s}\left( \mathbf{x}_{j,t} - \eta \nabla f_{j,t}(\mathbf{x}_{j,t}) \right) \right\| \\
= & \max_{(i,j)} \left\| \Pi_{\widehat{X}_i^s}\left( \mathbf{x}_{i,t} - \eta \nabla f_{i,t}(\mathbf{x}_{i,t}) \right) - \Pi_{\widehat{X}_j^s}\left( \mathbf{x}_{i,t} - \eta \nabla f_{i,t}(\mathbf{x}_{i,t}) \right) \right. \\
& \left. + \Pi_{\widehat{X}_j^s}\left( \mathbf{x}_{i,t} - \eta \nabla f_{i,t}(\mathbf{x}_{i,t}) \right) - \Pi_{\widehat{X}_j^s}\left( \mathbf{x}_{j,t} - \eta \nabla f_{j,t}(\mathbf{x}_{j,t}) \right) \right\| \\
\leq & \max_{(i,j)} \left\| \Pi_{\widehat{X}_i^s}\left( \mathbf{x}_{i,t} - \eta \nabla f_{i,t}(\mathbf{x}_{i,t}) \right) - \Pi_{\widehat{X}_j^s}\left( \mathbf{x}_{i,t} - \eta \nabla f_{i,t}(\mathbf{x}_{i,t}) \right) \right\| \\
& + \left\| \left( \mathbf{x}_{i,t} - \eta \nabla f_{i,t}(\mathbf{x}_{i,t}) \right) - \left( \mathbf{x}_{j,t} - \eta \nabla f_{j,t}(\mathbf{x}_{j,t}) \right) \right\| \\
\leq & O(\frac{1}{T^\rho}) + 2\left( (O(\frac{1}{T^\rho}) + 2\eta G)\frac{\sqrt{m}\beta}{1-\beta} \right) + 2G\eta = O(\frac{1}{T^\rho} + \eta).
\end{aligned}
\tag{42}
$$

The first inequality follows from the non-expansive property of projection, where $\left\|\Pi_{\mathcal{X}}(\mathbf{x}) - \Pi_{\mathcal{X}}(\mathbf{y})\right\| \leq \left\|\mathbf{x} - \mathbf{y}\right\|$ for any $\mathbf{x}, \mathbf{y}$ and a closed convex set $\mathcal{X}$, and the last inequality is based on Lemma 12, Lemma 14 and the Lipschitz continuity of the function sequence.

Substituting Equations (41) and (42) into Equation (40) and based on Lemma 15, we have

$$\left\|q(\mathbf{x}_{i,t+1}) - \mathbf{u}'_{i,t+1}\right\| \leq \left\|\sum_j [\mathbf{P}]_{ji} \mathbf{z}_{j,t} - \sum_j [\mathbf{P}]_{ji} q(\mathbf{y}_{j,t})\right\| + \left\|\sum_j [\mathbf{P}]_{ji} q(\mathbf{y}_{j,t}) - q(\sum_j [\mathbf{P}]_{ji} \mathbf{y}_{j,t})\right\|$$

$$\leq O(W^4 G_F^{3/2} \eta^{3/2}) + O(\frac{1}{T^{2\rho}} + \eta^2) = O(\frac{1}{T^{2\rho}} + \eta^{3/2}),$$

(43)

when $\eta$ is small enough. $\square$

*Proof of Theorem 5.* As for the proof of Theorem 3, we decompose the individual regret into three terms:

$$\sum_t \sum_i f_{i,t}(\mathbf{x}_{j,t}) - \sum_t f_t(\mathbf{x}_t^*) = \underbrace{\sum_{t=1}^{T_s-1} \sum_i f_{i,t}(\mathbf{x}_{j,t}) - f_{i,t}(\mathbf{x}_t^*)}_{\text{Term I}} + \underbrace{\sum_{t=T_s}^{T} \sum_i f_{i,t}(\mathbf{x}_{j,t}) - f_{i,t}(\tilde{\mathbf{x}}_t^*)}_{\text{Term II}} + \underbrace{\sum_{t=T_s}^{T} f_t(\tilde{\mathbf{x}}_t^*) - f_t(\mathbf{x}_t^*)}_{\text{Term III}},$$

(44)

where $\tilde{\mathbf{x}}_t^*$ is the projection of $\mathbf{x}_t^*$ on $\mathcal{X}_{\text{in}}^s$, which is a mutual subset of $\{\widehat{\mathcal{X}}_i^s\}_{i \in [m]}$ with $\tau_{\text{in}} = 2\mathcal{B}_r L$ based on Equation (22).

**The upper bound of Term I**:
Similar to the proof of convex part, during the estimation phase, $\gamma$ is less than $\frac{\Delta^s}{LL_A}$ to ensure the safeness of each agent's action, and based on the Lipschitz property we have

$$\sum_{t=1}^{T_s-1} \sum_i f_{i,t}(\mathbf{x}_{j,t}) - f_{i,t}(\mathbf{x}_t^*) = \sum_{t=1}^{T_s-1} \sum_i \tilde{f}_{i,t}\big(q(\mathbf{x}_{j,t})\big) - \tilde{f}_{i,t}\big(q(\mathbf{x}_t^*)\big)$$

$$\leq \sum_{t=1}^{T_s-1} \sum_i G_F W \left\|\mathbf{x}_{j,t} - \mathbf{x}_t^*\right\| \leq 2G_F W L m(T_0 + T_1).$$

(45)

**The upper bound of Term II**:
Define $\widehat{\mathcal{X}}_i^{s\prime} \triangleq \{q(\mathbf{x}) | \mathbf{x} \in \widehat{\mathcal{X}}_i^s\}$, (same for $\mathcal{X}_{\text{in}}^s$ and $\mathcal{X}^s$). Then, for any $q(\tilde{\mathbf{x}}_t^*) = \tilde{\mathbf{u}}_t^* \in \mathcal{X}_{\text{in}}^{s\prime}$, based on Equation (39), we

have

$$
\begin{aligned}
\eta\left(f_{i,t}(\mathbf{x}_{i,t}) - f_{i,t}(\tilde{\mathbf{x}}_t^*)\right) = &\; \eta\left(\tilde{f}_{i,t}(\mathbf{u}_{i,t}) - \tilde{f}_{i,t}(\tilde{\mathbf{u}}_t^*)\right) \\
\le &\; \eta\nabla\tilde{f}_{i,t}(\mathbf{u}_{i,t})^\top(\mathbf{u}_{i,t} - \tilde{\mathbf{u}}_t^*) \\
= &\; \left(\nabla\phi(\mathbf{u}_{i,t}) - \nabla\phi(\mathbf{z}_{i,t}) - \eta\nabla\tilde{f}_{i,t}(\mathbf{u}_{i,t})\right)^\top(\tilde{\mathbf{u}}_t^* - \mathbf{z}_{i,t}) \\
&+ \left(\nabla\phi(\mathbf{z}_{i,t}) - \nabla\phi(\mathbf{u}_{i,t})\right)^\top(\tilde{\mathbf{u}}_t^* - \mathbf{z}_{i,t}) + \eta\nabla\tilde{f}_{i,t}(\mathbf{u}_{i,t})^\top(\mathbf{u}_{i,t} - \mathbf{z}_{i,t}) \\
\le &\; \left(\nabla\phi(\mathbf{z}_{i,t}) - \nabla\phi(\mathbf{u}_{i,t})\right)^\top(\tilde{\mathbf{u}}_t^* - \mathbf{z}_{i,t}) + \eta\nabla\tilde{f}_{i,t}(\mathbf{u}_{i,t})^\top(\mathbf{u}_{i,t} - \mathbf{z}_{i,t}) \\
= &\; \mathcal{D}_\phi(\tilde{\mathbf{u}}_t^*, \mathbf{u}_{i,t}) - \mathcal{D}_\phi(\tilde{\mathbf{u}}_t^*, \mathbf{z}_{i,t}) - \mathcal{D}_\phi(\mathbf{z}_{i,t}, \mathbf{u}_{i,t}) + \eta\nabla\tilde{f}_{i,t}(\mathbf{u}_{i,t})^\top(\mathbf{u}_{i,t} - \mathbf{z}_{i,t}) \\
\le &\; \mathcal{D}_\phi(\tilde{\mathbf{u}}_t^*, \mathbf{u}_{i,t}) - \mathcal{D}_\phi(\tilde{\mathbf{u}}_t^*, \mathbf{z}_{i,t}) - \mathcal{D}_\phi(\mathbf{z}_{i,t}, \mathbf{u}_{i,t}) + \frac{1}{2}\left\|\mathbf{u}_{i,t} - \mathbf{z}_{i,t}\right\|^2 + \frac{\eta^2}{2}\left\|\nabla\tilde{f}_{i,t}(\mathbf{u}_{i,t})\right\|^2 \\
\le &\; \mathcal{D}_\phi(\tilde{\mathbf{u}}_t^*, \mathbf{u}_{i,t}) - \mathcal{D}_\phi(\tilde{\mathbf{u}}_t^*, \mathbf{z}_{i,t}) + \frac{\eta^2}{2}\left\|\nabla\tilde{f}_{i,t}(\mathbf{u}_{i,t})\right\|^2 \\
= &\; \mathcal{D}_\phi(\tilde{\mathbf{u}}_t^*, \mathbf{u}_{i,t}) - \mathcal{D}_\phi(\tilde{\mathbf{u}}_t^*, \mathbf{u}_{i,t}') + \mathcal{D}_\phi(\tilde{\mathbf{u}}_t^*, \mathbf{u}_{i,t}') - \mathcal{D}_\phi(\tilde{\mathbf{u}}_t^*, \mathbf{z}_{i,t}) + \frac{\eta^2}{2}\left\|\nabla\tilde{f}_{i,t}(\mathbf{u}_{i,t})\right\|^2 \\
\le &\; \mathcal{D}_\phi(\tilde{\mathbf{u}}_t^*, \mathbf{u}_{i,t}) - \mathcal{D}_\phi(\tilde{\mathbf{u}}_t^*, \mathbf{u}_{i,t}') + \sum_j[\mathbf{P}]_{ji}\mathcal{D}_\phi(\tilde{\mathbf{u}}_t^*, \mathbf{z}_{j,t-1}) - \mathcal{D}_\phi(\tilde{\mathbf{u}}_t^*, \mathbf{z}_{i,t}) + \frac{\eta^2}{2}\left\|\nabla\tilde{f}_{i,t}(\mathbf{u}_{i,t})\right\|^2,
\end{aligned}
\tag{46}
$$

where the second inequality is based on the optimality of $\mathbf{z}_{i,t}$; the fourth inequality is due to the strong convexity of $\phi(\cdot)$ and the fifth inequality is based on Assumption 9.

Based on Theorem 16, Lemma 17, and the Lipschitz assumption on $\mathcal{D}_\phi$, we have

$$
\left\|\mathcal{D}_\phi(\tilde{\mathbf{u}}_t^*, \mathbf{u}_{i,t}) - \mathcal{D}_\phi(\tilde{\mathbf{u}}_t^*, \mathbf{u}_{i,t}')\right\| \le W\left\|\mathbf{u}_{i,t} - \mathbf{u}_{i,t}'\right\| \le O\left(W\left(\frac{1}{T^{2\rho}} + \eta^{3/2}\right)\right).
\tag{47}
$$

And based on Lemma 14, we get

$$
\max_{i,j\in[m]}\left\|\mathbf{u}_{i,t} - \mathbf{u}_{j,t}\right\| = \max_{i,j\in[m]}\left\|q(\mathbf{x}_{i,t}) - q(\mathbf{x}_{j,t})\right\| = O(W\eta).
\tag{48}
$$

With Equations (46), (47) and (48), we derive

$$
\begin{aligned}
\tilde{f}_{i,t}(\mathbf{u}_{j,t}) - \tilde{f}_{i,t}(\tilde{\mathbf{u}}_t^*) = &\; \tilde{f}_{i,t}(\mathbf{u}_{j,t}) - \tilde{f}_{i,t}(\mathbf{u}_{i,t}) + \tilde{f}_{i,t}(\mathbf{u}_{i,t}) - \tilde{f}_{i,t}(\tilde{\mathbf{u}}_t^*) \\
\le &\; G_F\left\|\mathbf{u}_{i,t} - \mathbf{u}_{j,t}\right\| + O\left(W\left(\frac{1}{\eta T^{2\rho}} + \eta^{1/2}\right)\right) \\
&+ \frac{1}{\eta}\sum_j[\mathbf{P}]_{ji}\mathcal{D}_\phi(\tilde{\mathbf{u}}_t^*, \mathbf{z}_{j,t-1}) - \frac{1}{\eta}\mathcal{D}_\phi(\tilde{\mathbf{u}}_t^*, \mathbf{z}_{i,t}) + \frac{\eta}{2}\left\|\nabla\tilde{f}_{i,t}(\mathbf{u}_{i,t})\right\|^2 \\
\le &\; O(G_F W\eta) + O\left(W\left(\frac{1}{\eta T^{2\rho}} + \eta^{1/2}\right)\right) \\
&+ \frac{1}{\eta}\sum_j[\mathbf{P}]_{ji}\mathcal{D}_\phi(\tilde{\mathbf{u}}_t^*, \mathbf{z}_{j,t-1}) - \frac{1}{\eta}\mathcal{D}_\phi(\tilde{\mathbf{u}}_t^*, \mathbf{z}_{i,t}) + \frac{\eta}{2}\left\|\nabla\tilde{f}_{i,t}(\mathbf{u}_{i,t})\right\|^2.
\end{aligned}
\tag{49}
$$

Based on the definition of Bregman divergence, we have the following relationship

$$
\begin{aligned}
&\mathcal{D}_\phi(\tilde{\mathbf{u}}_t^*, \mathbf{z}_{i,t-1}) - \mathcal{D}_\phi(\tilde{\mathbf{u}}_t^*, \mathbf{z}_{i,t}) \\
= &\; \left(\nabla\phi(\mathbf{z}_{i,t}) - \nabla\phi(\mathbf{z}_{i,t-1})\right)^\top(\tilde{\mathbf{u}}_t^* - \mathbf{z}_{i,t}) + \mathcal{D}_\phi(\mathbf{z}_{i,t}, \mathbf{z}_{i,t-1}) \\
= &\; \left(\nabla\phi(\mathbf{z}_{i,t}) - \nabla\phi(\mathbf{z}_{i,t-1})\right)^\top\tilde{\mathbf{u}}_t^* + \left(\phi(\mathbf{z}_{i,t}) - \nabla\phi(\mathbf{z}_{i,t})^\top\mathbf{z}_{i,t}\right) - \left(\phi(\mathbf{z}_{i,t-1}) - \nabla\phi(\mathbf{z}_{i,t-1})^\top\mathbf{z}_{i,t-1}\right).
\end{aligned}
\tag{50}
$$

Summing Equation (49) over $i$, based on Equation (50) we get

$$
\sum_i \tilde{f}_{i,t}(\mathbf{u}_{j,t}) - \tilde{f}_{i,t}(\tilde{\mathbf{u}}_t^*)
$$

$$
\leq O\big(mG_F W\eta\big) + O\big(mW(\frac{1}{\eta T^{2\rho}} + \eta^{1/2})\big) + \sum_i \frac{\eta}{2}\big\|\nabla\tilde{f}_{i,t}(\mathbf{u}_{i,t})\big\|^2 \tag{51}
$$

$$
+\frac{1}{\eta}\sum_i \Big[ (\nabla\phi(\mathbf{z}_{i,t}) - \nabla\phi(\mathbf{z}_{i,t-1}))^\top \tilde{\mathbf{u}}_t^* + \big(\phi(\mathbf{z}_{i,t}) - \nabla\phi(\mathbf{z}_{i,t})^\top\mathbf{z}_{i,t}\big) - \big(\phi(\mathbf{z}_{i,t-1}) - \nabla\phi(\mathbf{z}_{i,t-1})^\top\mathbf{z}_{i,t-1}\big) \Big].
$$

Then, by summing Equation (51) over $[T_s, \ldots, T]$, we have

$$
\sum_{t=T_s}^{T}\sum_i \tilde{f}_{i,t}(\mathbf{u}_{j,t}) - \tilde{f}_{i,t}(\tilde{\mathbf{u}}_t^*)
$$

$$
\leq O\big(mTG_F W\eta\big) + O\big(mTW(\frac{1}{\eta T^{2\rho}} + \eta^{1/2})\big) + \sum_{t=T_s}^{T}\sum_i \frac{\eta}{2}\big\|\nabla\tilde{f}_{i,t}(\mathbf{u}_{i,t})\big\|^2 \tag{52}
$$

$$
+\frac{1}{\eta}\Bigg[ \sum_{t=T_s-1}^{T-1}\sum_i (\tilde{\mathbf{u}}_t^* - \tilde{\mathbf{u}}_{t+1}^*)^\top\nabla\phi(\mathbf{z}_{i,t}) + \sum_i \nabla\phi(\mathbf{z}_{i,T})^\top\tilde{\mathbf{u}}_T^* - \sum_i \nabla\phi(\mathbf{z}_{i,T_s-1})^\top\tilde{\mathbf{u}}_{T_s-1}^* \Bigg]
$$

$$
+\frac{1}{\eta}\sum_i \Big[ \big(\phi(\mathbf{z}_{i,T}) - \nabla\phi(\mathbf{z}_{i,T})^\top\mathbf{z}_{i,T}\big) - \big(\phi(\mathbf{z}_{i,T_s-1}) - \nabla\phi(\mathbf{z}_{i,T_s-1})^\top\mathbf{z}_{i,T_s-1}\big) \Big].
$$

**The upper bound of Term III**:
Based on Lemma 9, we have for any $\mathbf{x}_t^* \in \mathcal{X}^s$ and its projection to $\mathcal{X}_{\text{in}}^s$: $\tilde{\mathbf{x}}_t^*$

$$
\sum_{t=T_s}^{T}\sum_i \big(\tilde{f}_{i,t}(q(\tilde{\mathbf{x}}_t^*)) - \tilde{f}_{i,t}(q(\mathbf{x}_t^*))\big) \leq \sum_{t=T_s}^{T}\sum_i G_F W\big\|\tilde{\mathbf{x}}_t^* - \mathbf{x}_t^*\big\| \leq mTG_F W\frac{2\sqrt{d}L\mathcal{B}_r}{C(\mathbf{A},\mathbf{b})}. \tag{53}
$$

Substituting Equations (45), (52) and (53) into Equation (44), the final regret bound is as

$$
\sum_{t=1}^{T}\sum_i \big(f_{i,t}(\mathbf{x}_{j,t}) - f_{i,t}(\mathbf{x}_t^*)\big)
$$

$$
\leq O\big(mTG_F W\eta\big) + O\big(mTW(\frac{1}{\eta T^{2\rho}} + \eta^{1/2})\big) + \sum_{t=T_s}^{T}\sum_i \frac{\eta}{2}\big\|\nabla\tilde{f}_{i,t}(\mathbf{u}_{i,t})\big\|^2
$$

$$
+\frac{1}{\eta}\Bigg[ \sum_{t=T_s-1}^{T-1}\sum_i (\tilde{\mathbf{u}}_t^* - \tilde{\mathbf{u}}_{t+1}^*)^\top\nabla\phi(\mathbf{z}_{i,t}) + \sum_i \nabla\phi(\mathbf{z}_{i,T})^\top\tilde{\mathbf{u}}_T^* - \sum_i \nabla\phi(\mathbf{z}_{i,T_s-1})^\top\tilde{\mathbf{u}}_{T_s-1}^* \Bigg] + 2G_F WLm(T_0 + T_1)
$$

$$
+\frac{1}{\eta}\sum_i \Big[ \big(\phi(\mathbf{z}_{i,T}) - \nabla\phi(\mathbf{z}_{i,T})^\top\mathbf{z}_{i,T}\big) - \big(\phi(\mathbf{z}_{i,T_s-1}) - \nabla\phi(\mathbf{z}_{i,T_s-1})^\top\mathbf{z}_{i,T_s-1}\big) \Big] + mTG_F W\frac{2\sqrt{d}L\mathcal{B}_r}{C(\mathbf{A},\mathbf{b})}
$$

$$
= O(T_0 + T_1 + T\sqrt{\eta} + \frac{T\sqrt{\log T_0}}{\sqrt{T_0}} + \frac{1}{\eta} + \frac{1}{\eta}\sum_{t=T_s}^{T}\big\|\tilde{\mathbf{u}}_t^* - \tilde{\mathbf{u}}_{t+1}^*\big\|), \tag{54}
$$

where the final regret bound is proved by applying the specified $\eta$ and $T_0$. By choosing $\rho$ as a large enough number, $\frac{1}{\eta T^{2\rho}}$ is dominated by $\eta^{1/2}$. $\qquad\square$

