# OpenReview forum: "Dynamic Regret Analysis of Safe Distributed Online Optimization for Convex and Non-convex Problems"
_TMLR — Accepted by TMLR_

### Review · Reviewer_dTFm · 2023-06-11

**Summary Of Contributions:**

This paper investigates the problem of safe distributed online optimization, in which all local learners need to satisfy an unknown set of linear safety constraints. Both convex functions and a certain type of non-convex functions are considered, and an algorithm called D-Safe-OGD is proposed. Moreover, the authors prove that it can attain a dynamic regret bound of $O(T^{2/3}\sqrt{\log T}+T^{1/3}C_T^\ast)$ for convex functions, and a dynamic regret bound of $O(T^{2/3}\sqrt{\log T}+T^{2/3}C_T^\ast)$ for the certain type of non-convex functions.

**Audience:**

Yes

**Broader Impact Concerns:**

I have no concerns about the ethical implications of the work.

**Claims And Evidence:**

Yes

**Requested Changes:**

1) The authors should provide a more formalized introduction to Theorem 3.1 in Bonnans et al. (1998) and Theorem 3.7 in Shi et al. (2015) before using them.
2) In the first inequality of Eq. (41), the authors have utilized $\\|\Pi_{X}(x)-\Pi_{X}(y)\\|\leq\\|x-y\\|$ for any $x,y$, which is not very straightforward. So, it would be better if the authors could provide detailed explanations.
3) In page 3, Besbes et al. (2015) only establish an $O(\sqrt{T(1+V_T)}\log T)$ dynamic regret bound for strongly convex functions.
4) Lemma 1 is almost the same as Lemma 1 in Chaudhary & Kalathil (2022). So, the authors should also give some credit to Chaudhary & Kalathil (2022) when introducing this lemma.
5) The upper bounds presented by this paper ignore all other constants except $T$, $\delta$, and $C_T^\ast$. It would be better if the authors could also provide these omitted constants.
6) When discussing the related work on dynamic regret, the authors should clearly distinguish the general dynamic regret with respect to any sequence of comparators and the worst-case one studied in this paper.
7) Some related studies on the worse-case dynamic regret [1][2][3] are missed, and should also be discussed.
8) Some related studies on distributed online optimization with complex constraints [5][6] are missed, and should also be discussed.

[1] Zhang et al. Dynamic Regret of Strongly Adaptive Methods. In ICML, 2018.

[2] Wan et al. Projection-Free Online Learning in Dynamic Environments. In AAAI, 2021.

[3] Wan et al. Improved Dynamic Regret for Online Frank-Wolfe. In arXiv:2302.05620, 2023.

[4] Zhang et al. Projection-free Distributed Online Learning in Networks. In ICML, 2017.

[5] Wan et al. Projection-free Distributed Online Convex Optimization with $O(\sqrt{T})$ Communication Complexity. In ICML, 2020.

**Strengths And Weaknesses:**

# Strengths
1) To the best of my knowledge, this paper is the first work to study the problem of safe distributed online optimization.
2) Compared with existing studies on centralized safe online optimization, this paper provides a more careful analysis to handle the difference of safe sets estimated by these local learners.
3) The dynamic regret analysis and the distributed extension of the existing non-convex online optimization are new to me.

# Weaknesses
1) The main idea to handle the unknown safety constraints almost is the same as that of Chaudhary & Kalathil (2022), though they only consider the centralized setting. It seems that the distributed setting does not bring some significant challenges for satisfying the unknown safety constraints, which makes this paper incremental to some extent.
2) This paper only provides the upper bounds on dynamic regret. It is not clear whether these bounds are optimal or not.
3) In the theoretical analysis, the authors have utilized some useful conclusions from previous studies, but the introduction about these results is not easy for readers to check, e.g., the application of Theorem 3.1 in Bonnans et al. (1998) (below Eq. (27) in this submission), the application of Theorem 3.7 in Shi et al. (2015) (below Eq. (17) in this submission).

Moreover, as listed below, this paper needs to make some important modifications, and the current version cannot be accepted.

---

> ### Author Response · Authors · 2023-07-06
>
> We appreciate the comments and constructive feedback from the reviewer.
> >W
>
> Please note that in contrast to the centralized safe online learning in (Chaudhary \& Kalathil (2022)), as the constraint parameters of each agent are estimated in a distributed manner in our work, the problem ends up being distributed online learning with **different** feasible sets among agents. This problem is far from trivial and cannot be handled with slight adjustments of the previous work. Therefore, we must quantify the effect of projections on different estimated feasible sets. This error quantification plays a key role in characterizing the final regret bound. One main contribution of this work is to tackle this problem using the sensitivity analysis in Bonnans et al.(1998) (please refer to Lemma 12 for detailed information).
>
> In what follows, we provide the answers to the requested changes:
> >RC1
>
> Thank you for this suggestion. We have included formalized statements of these two theorems in the revised manuscript. Please see Theorems 10 and 13, highlighted in blue.
> >RC2
>
> We addressed this concern by mentioning that the projection operator is non-expansive for a closed convex set. Based on the reviewer's suggestion, we added this explanation in the revised manuscript. Please see the highlighted blue text after Equation (42).
> >RC3
>
> We thank the reviewer for pointing this out. We fixed this issue in the revised manuscript.
> >RC4
>
> Thanks for this suggestion. We modified Lemma 1 accordingly.
> >RC5
>
> We thank the reviewer for sharing this point of view. In order to clearly present our main results and deliver the key message to the readers, we chose to write the bounds in theorems in terms of the time horizon $T$, the path length $C^*_T$, the network properties (i.e., $\beta$), and the probability parameter $\delta$. The exact bounds with all of the problem dependent constants are rather messy and might distract the readers from the key parameters. Therefore, to address the reviewer's concern, we have now added explanations after Theorems 3 and 5 to refer the interested readers to exact bounds in the Appendix.
> >RC6
>
>  To avoid confusion, as the reviewer suggested, we now added the definition of regret with respect to general comparator sequence in Equation (5) and clarified the related works focusing on this metric.
> >RC7
>
> Thank you for pointing out these references. We included a discussion of these related works around Equation (6). Please see the blue highlighted text in the revised manuscript.
> >RC8
>
>  We also added the discussion of these references in page 4. Please refer to the blue highlighted text.

---

### Review · Reviewer_YPwj · 2023-06-12

**Summary Of Contributions:**

The paper studies safe distributed online optimization. In particular, a network of agents optimizes a global, time-varying function subject to linear unknown constraints. Each agent observes only local information about the objective function and can communicate with their neighbors. Differently from the objective function, the linear safety constraints are common to all the agents. The authors provide an algorithm that provides a $O(T^{2/3} \sqrt{\log T}+ T^{1/3} C^*_T))$ dynamic regret bound for convex functions, where $C_T^*$ denotes the path length of the best minimizer sequence. Moreover, they provide an algorithm for some non-convex problems.


**Audience:**

Yes

**Broader Impact Concerns:**

no concerns

**Claims And Evidence:**

Yes

**Requested Changes:**

Please highlight the technical challenges you face to extend distributed algorithms with known constraints to your setting.


**Strengths And Weaknesses:**

This work provides the first algorithm for distributed safe optimization that guarantees that the constraints are not violated at any round with high probability. However, from the technical point of view, it is not clear which are the main contributions of the paper since most of the results are heavily based on previous work.

Regarding Algorithm 1. The use of a convex combination between a known safe action and a zero-mean random vector is not novel, and the second part of the parameter estimation algorithm uses a known algorithm.

Regarding Algorithm 2. The analysis seems quite close to previous work, once you notice the existence of a mutual shrunk polytope.

Regarding the non-convex setting. The main goal of the study of this setting is to understand why online gradient descent works well in some specific non-convex settings. It is not clear which is the significance of extending the analysis to distributed algorithms with unknown safety constraints.

---

> ### Author Response · Authors · 2023-07-06
>
> $\newcommand{\norm}[1]{\left\lVert#1\right\rVert}$
> $\newcommand{\Ab}{\mathbf{A}}$
> $\newcommand{\Ib}{\mathbf{I}}$
> $\newcommand{\Pb}{\mathbf{P}}$
> $\newcommand{\xb}{\mathbf{x}}$
> $\newcommand{\ub}{\mathbf{u}}$
> $\newcommand{\xbhat}{\hat{\mathbf{x}}}$
> $\newcommand{\xbtop}{\xb^{\top}_{i,t}}$
> We thank the reviewer for carefully reading our manuscript and providing thoughtful comments.
> > W & RC
>
> While we agree that Algorithm 1 leverages EXTRA, the estimation part only serves as a building block in our work, and it is not our main contribution. To be specific, the estimation of the unknown safe set causes further challenges for analyzing the dynamic regret, characterizing which is our main contribution (both in the convex and non-convex settings). Therefore, we would like to emphasize that even the analysis of the convex setting depends on a key lemma (Lemma 12 in the revised manuscript), and as such, we respectfully disagree that previous work (with slight adjustments) could answer the convex setting. We outline the main technical challenges that we tackle in this work as follows:
>
> (1) Compared to the setup where the constraint parameters are known, for the **unknown** setup since each agent constructs its feasible set based on its own estimate, the problem is transformed into a *distributed* online learning with **different** feasible sets across agents. Therefore, unlike distributed online optimization on a *common known* set, we must quantify the effect of projections on different estimated feasible sets. This error quantification plays a key role in characterizing the final regret bound. We tackle this issue by showing that the projections on different estimated feasible sets can be cast as a *perturbation analysis*  of a semi-definite optimization, which allows us to leverage the results of sensitivity analysis in Bonnans et al.(1998) to get a desirable upper bound. Please refer to Lemma 12 for detailed information, where we quantify the distance between projections of a point onto two sets that are *close enough*. Note that this analysis is even required for the convex setting.
>
> (2) For the non-convex case, Ghai et al. (2022) provided a sub-linear regret bound by showing that OGD applied to non-convex problems can be suitably approximated by OMD applied to the corresponding reparameterized convex problems (through a non-linear mapping $q(\cdot)$). Specifically, if OGD starts from a point $\xb_t$ and OMD starts from $\ub_t = q(\xb_t)$, then $\norm{\ub_{t+1} - q(\xb_{t+1})}$, the difference after updates, can be bounded by $O(\eta^{3/2})$ ($\eta$ being the step size), which is small enough to achieve a sub-linear regret bound. However, in addition to the challenge discussed in the previous paragraph (i.e., Lemma 12), this result is not readily applicable in the **distributed** setup due to the communication step. We handled this problem by considering the approximation of the mapping $q(\cdot)$. We related the difference between the  iterates of distributed OGD and distributed OMD to the bound on the difference between projections on different estimated sets to tackle this challenge. Please refer to Lemma 17 for the detailed analysis.
>
> (3) As opposed to the closely related works Ghai et al. (2022) and Chaudhary & Kalathil (2022), we characterize the performance of D-safe-OGD using **dynamic** regret, which subsumes the corresponding theoretical results (static regret bounds) in those works.
>
> We have discussed these points in Section 1.2 (Contributions). Please note that we outlined only the key technical challenges here. Together with other results of this paper, we believe our work provides novel theoretical contributions to the field of online optimization.

---

### Review · Reviewer_J4zV · 2023-06-22

**Summary Of Contributions:**

This paper studies safe distributed online optimization over an unknown set of linear constraints. Dynamic regrets for convex and non-convex loss functions have been established. Overall the topic of the paper is interesting and the paper is well written.

**Audience:**

Yes

**Broader Impact Concerns:**

No concerns

**Claims And Evidence:**

Yes

**Requested Changes:**

1. It is suggested to provide the explicit expression of $\nabla l_{i}(\widehat{A}_{i}^{t}) $ in Algorithm 1 and perform a detailed analysis of the computational complexity of Algorithm 1.
2. Please specify the choice of the step size $\alpha$ in Lemma 2.
3. The learning rate $\eta$ in Algorithm 2 requires the information of $T$, which is global, is it possible to adopt adaptive learning rate that depends only on $t$?
4. The authors should specify what type of non-convexity is considered in Section 6. In addition, why is the definition of the dynamic regret identical to that of the convex setting?
5. Does the safe baseline action in Assumption 3 affect the accuracy of the parameter estimation? I think more lines should be added to discuss about this assumption.

**Strengths And Weaknesses:**

The constraint information is unknown, which is different from most of the existing literature on online distributed optimization under inequality constraints.

The regret bounds are in dynamic setting and match those of the centralized counterpart.

Some details regarding the computational complexity, learning rate, and non-convexity need to further clarified.

---

> ### Author Response · Authors · 2023-07-06
>
> $\newcommand{\norm}[1]{\left\lVert#1\right\rVert}$
> $\newcommand{\Ab}{\mathbf{A}}$
> $\newcommand{\Ib}{\mathbf{I}}$
> $\newcommand{\Pb}{\mathbf{P}}$
> $\newcommand{\xb}{\mathbf{x}}$
> $\newcommand{\ub}{\mathbf{u}}$
> $\newcommand{\xbhat}{\hat{\mathbf{x}}}$
> $\newcommand{\xbtop}{\xb^{\top}_{i,t}}$
> We thank the reviewer for the constructive feedback and provide the following answers to the requested changes:
> >RC1
>
> Given that $l_i(\Ab)=\sum_{t=1}^{T_0}\norm{\Ab\xb_{i,t} - \hat{\xb}_{i,t}}^2 + \frac{\lambda}{m}\norm{\Ab}_F^2$, the gradient takes the following form
>
> $$
> \nabla l_i(\Ab) = \sum_{t=1}^{T_0}\left[2\Ab\xb_{i,t} \xbtop  -  2\xbhat_{i,t} \xbtop\right] + \frac{2\lambda}{m}\Ab,
> $$
>
> where $\Ab \in \mathrm{R}^{n\times d}$ and $\xb_{i,t}, \hat{\xb}_{i,t}\in \mathrm{R}^d$. Based on the expression above, each gradient computation requires $O(T_0 nd)$ operations. Therefore, for the optimization part of Algorithm 1 (EXTRA), the computation cost is $O\big(mT_1(mnd + T_0 nd)\big)$, where the $mnd$ term accounts for the weighted sum in line 15 of the algorithm, and the $mT_1$ term appears due to the fact that the algorithm runs for $T_1$ iterations and $m$ agents. Therefore, the total computation cost of Algorithm 1 is $O\big(mT_1(mnd + T_0 nd)\big)$ noting that the computation cost of $O(mdT_0)$ in the data collection phase is dominated by the cost of running EXTRA. As the reviewer suggested, we added this computational cost analysis in the revised manuscript after Lemma 2
> >RC2
>
> The step size $\alpha$ is chosen based on Theorem 3.7 in Shi et al.(2015), such that $\alpha < \frac{2\mu_g\lambda_{min}(\tilde{\Pb})}{L_f^2}$, where $\mu_g$ and $L_f$ characterize the strong convexity and Lipschiz smoothness of the finite sum problem $\sum_{i=1}^m l_i(\Ab_i)$, respectively. In our case, based on the definition of $l_i(\Ab)$, it is straightforward to verify strong convexity and Lipschitz smoothness as follows
>
> $$
> 2\frac{\lambda}{m}\Ib\preceq\nabla^2 l_i(\Ab) = \sum_{t=1}^{T_0} 2\begin{bmatrix}\xb_{i,t}\xb_{i,t}^{\top} & & & \\\\ & \xb_{i,t}\xb_{i,t}^{\top} & & \\\\ & & \ddots & \\\\  & & & \xb_{i,t}\xb_{i,t}^{\top}\end{bmatrix} + 2\frac{\lambda}{m}\Ib \preceq 2(T_0 L^2 + \frac{\lambda}{m})\Ib,
> $$
>
> from which we can select any $\alpha < \frac{(\lambda/m)\lambda_{min}(\tilde{\Pb})}{(T_0L^2 + \frac{\lambda}{m})^2}$ to achieve the geometric convergence of EXTRA algorithm. We included a brief discussion in this regard in the modified manuscript after Equation (18).
> >RC3
>
> Yes, it is possible to remove the dependence to $T$. One standard way is to use the *doubling trick* (see e.g., Section 2.3.1 in [1] or Section 2.3 in [2]), which is a well-known approach to make an online algorithm achieve the same regret bound (in terms of order) without the knowledge of $T$. The rough idea is to divide the entire horizon into several intervals that form a geometric sequence, and in each interval, the algorithm is run with the knowledge of the length of that interval. Then, by accumulating the regret in all intervals, it can be shown that the total regret is as of the same regret order of the algorithm that is run with the knowledge of $T$.
>
> >RC4
>
> We are inspired by the non-convex setting discussed in Ghai et al. (2022), where the non-convex functions have geometric properties characterized in Assumptions 5,6 and 7. This allows a non-convex function $f_t(\xb)$ to be reparameterized as a convex function $\tilde{f}_t(\ub)$ through a mapping $\ub = q(\xb)$ with some smoothness properties. As shown in Ghai et al. (2022), this class of non-convex functions allows us to analyze the performance of online gradient descent (OGD) through approximating it as online mirror descent (OMD) applied on the reparameterized convex functions, from which we can get meaningful results even when we consider the regret measure in the *convex* setting. We extend this equivalence to “distributed” variants of OGD and OMD under the additional complexity that the constraint set is unknown, and it can only be approximated via Algorithm 1. Our focus is on analyzing the effect of (i) the constraint estimation as well as (ii) the distributed setup in non-convex online learning, and we also generalize the analysis of Ghai et al. (2022) to the dynamic regret. We mentioned in the introduction (item 3 of contributions) that we only focus on certain non-convex geometries, but we deferred the technical assumptions to Section 6 to enhance the readability of the manuscript.
> >RC5
>
> This is a great point. As the applied action is defined as $\xb_{i,t}= (1-\gamma)\xb^s + \gamma\zeta_{i,t}$, where $\gamma = \frac{\Delta^s}{LL_A}$, the safety gap $\Delta^s$ of the safe baseline action affects the weight $\gamma$ in the linear combination. Also, we can see from Lemma 2 that if $\gamma$ is larger, we have a tighter error bound for the estimation. In the revised manuscript (after Lemma 2), we have included the discussion of the safe baseline action role in the estimation.

---

> > ### Author Response · Authors · 2023-07-06
> > **References**
> >
> > >References
> >
> > [1] S. Shalev-Shwartz et al., “Online learning and online convex optimization,” Foundations and Trends® in Machine Learning,
> > vol. 4, no. 2, pp. 107–194, 2012.
> >
> > [2] N. Cesa-Bianchi and G. Lugosi, Prediction, learning, and games. Cambridge university press, 2006.

---

### Decision · Action_Editors · 2023-08-19

**Recommendation:** Accept with minor revision

**Comment:**

The problem being studied and the technical results are new. The majority of the reviewers found the results of interest. One expert reviewer has concerns about the "relatively weak technical contribution" of the work as most of the techniques used for the algorithm and its analysis are from the existing work. The authors highlighted the challenges they had to overcome but the reviewer's concern on significance remains.

As per TMLR evaluation criteria https://jmlr.org/tmlr/editorial-policies.html#evaluation, the work meets both the "soundness" rule and the "interest" rule.   The "interest" is clear from the positive subset of the reviewers and I went through the paper with interest too. Overall, I am happy to recommend the paper for publication at TMLR.

I have some more comments for the authors consideration as a minor revision

1. The optimality in each parameter of the problem should be discussed more clearly.  The authors are frank that the regret bound they obtained might not be optimal in $C_T^*$. But is the $T^{1/3}$ term from Theorem 3 optimal (when the third term dominates)?  I am not sure if the conjecture that the optimal dependence on $C_T^*$ should be $\sqrt{C_T^*}$ in Remark 4 makes sense because when $C_T^* = T$ the expression is superlinear.

2. I encourage the authors to add a "technical summary" after the summary of results at the end of the introduction, to comment on the proof techniques (which are new from this paper, which ones are adapted from existing work, and which ones are borrowed directly from existing work). An upfront discussion of the techniques helps the theoretical audience to quickly understand the "meat" of the work and determine whether some of the techniques from this paper can be useful for them elsewhere.

3. The literature review on dynamic regret is a bit messy.  It introduced too many different versions of "path lengths" and different settings.  It might make sense to categorize the discussion a bit, and also to defer very involved discussion to the end of the paper (or even in the appendix).

For example, the dynamic regret guarantee for the pointwise minimizer sequence C* is very different from (and much weaker than) that of the universal dynamic regret that competes with all sequences (Universal dynamic regret). Recent work has obtained optimal universal dynamic regret bounds for strongly convex / exp-concave losses (e.g., Baby and Wang, COLT'2021; AISTATS'2022) which is qualitatively very different compared to those of O(C_T^*) from Mokhtari et al. (2016). Generally speaking, in almost all stochastic / adversarial cases $C_T^* = O(T)$, e.g., linear regression, logistic regression, universal portfolio... the comparator sequence is not instantiated as pointwise optimal sequence, but rather another more slowly changing sequence for the interest of a better bias/variance tradeoff.

 Related to the above, the authors also introduced many other variants of pathlength:  V_T,  C'_T, and squared variants. These are all good but dumping them on the readers when they are still trying understand what *this paper* is about is a bit distracting.  This is especially so because none of these alternative pathlengths were used in this paper.    Usually such literature discussion should focus on work that is directly relevant to the current work, e.g. either those that provide lower bounds for this setting, or those that the authors have borrowed settings / techniques / algorithmic ideas.  A longer discussion of the related work can be deferred to the end after the main results are presented.

**Audience:**

Yes. The targeted audience would be a combination of online learning theorists,  distributed /federated optimization researchers, and potentially some applied science researchers down the road.

**Claims And Evidence:**

The paper studied dynamic regret (w.r.t. the pointwise optimal sequence, rather than the stronger "universal" counterpart)  in the distributed setting with (unknown) safety constraints.

The main technical claim is a regret upper bound in this setting for general convex losses, and certain specialized families of non-convex losses. Proofs are provided and reviewers have checked the correctness of the proof to the best of their ability.  I found the results believable too.

---

> ### Author Response · Authors · 2023-09-18
>
> We thank the action editor (AE) for providing constructive feedback on our manuscript. We have modified the paper as follows
>
> > Comment 1
>
> The original regret provided in Remark 4 was not optimal as the dependence of $\eta$ on $T$ was not exactly optimized. We fixed this issue and provided the accurate regret bound in Remark 4. The new bound is not superlinear in $T$ even in the worst-case.
>
> > Comment 2
>
> As the AE suggested, at the bottom of Section 1, we added a summary of the technical analysis (Section 1.3) to highlight our technical contributions and describe a road-map of how we combine the previous results to achieve the results in the current work.
>
> > Comment 3
>
> We agree with the AE that the discussion on $V_T, D_T$ and $C'_T$, which are less relevant to our results, could be deferred to later parts of the manuscript. We discussed those parts in a new section (Section 7) before the conclusion. We also complemented the discussion of universal dynamic regret bounds (in terms of $C_T$) and included the new references (Baby and Wang, COLT'2021; AISTATS'2022) suggested by the AE. This discussion appears right after equation (5).